# An Adaptive Parameter Optimization Deep Learning Model for Energetic Liquid Vision Recognition Based on Feedback Mechanism

**DOI:** 10.3390/s24206733

**Published:** 2024-10-19

**Authors:** Lu Chen, Yuhao Yang, Tianci Wu, Chiang Liu, Yang Li, Jie Tan, Weizhong Qian, Liang Yang, Yue Xiu, Gun Li

**Affiliations:** 1School of Aeronautics and Astronautics, University of Electronic Science and Technology of China, Chengdu 611731, China; lchen@std.uestc.edu.cn (L.C.); yangyh@std.uestc.edu.cn (Y.Y.); tcwu@std.uestc.edu.cn (T.W.); liuchianguestc@163.com (C.L.); yli@std.uestc.edu.cn (Y.L.); tanjie@std.uestc.edu.cn (J.T.); wzqian@uestc.edu.cn (W.Q.); lynn_workshop@163.com (L.Y.); 2School of Air Traffic Management, Civil Aviation Flight University of China, Deyang 618307, China; xiuyue12345678@163.com

**Keywords:** energetic liquid, viscosity visual recognition, integrated deep learning model, deep genetic feedback, Adaptive Genetic Selector

## Abstract

The precise detection of liquid flow and viscosity is a crucial challenge in industrial processes and environmental monitoring due to the variety of liquid samples and the complex reflective properties of energetic liquids. Traditional methods often struggle to maintain accuracy under such conditions. This study addresses the complexity arising from sample diversity and the reflective properties of energetic liquids by introducing a novel model based on computer vision and deep learning. We propose the DBN-AGS-FLSS, an integrated deep learning model for high-precision, real-time liquid surface pointer detection. The model combines Deep Belief Networks (DBN), Feedback Least-Squares SVM classifiers (FLSS), and Adaptive Genetic Selectors (AGS). Enhanced by bilateral filtering and adaptive contrast enhancement algorithms, the model significantly improves image clarity and detection accuracy. The use of a feedback mechanism for reverse judgment dynamically optimizes model parameters, enhancing system accuracy and robustness. The model achieved an accuracy, precision, F1 score, and recall of 99.37%, 99.36%, 99.16%, and 99.36%, respectively, with an inference speed of only 1.5 ms/frame. Experimental results demonstrate the model’s superior performance across various complex detection scenarios, validating its practicality and reliability. This study opens new avenues for industrial applications, especially in real-time monitoring and automated systems, and provides valuable reference for future advancements in computer vision-based detection technologies.

## 1. Introduction

With the continuous advancement of science and technology and the increasing demand for production, the detection of liquid flow and viscosity has become particularly important. In scientific research, liquid flow measurement has wide-ranging applications. It plays a crucial role in studying environmental issues such as water pollution and soil infiltration [1]. Additionally, many new measurement techniques are continuously emerging to meet the growing needs in this field [2,3].

The method of measuring liquid flow and viscosity by visually detecting the upper and lower pointers and calculating the time difference presents several advantages. While the diversity of test liquids adds complexity to detection, traditional visual inspection often struggles with varying viscosities and densities, leading to inconsistencies and errors. In contrast, computer vision technology can adapt flexibly to the characteristics of energetic liquids, ensuring more accurate measurements [4]. A unified visual detection system eliminates the need for mass measurement to determine fixed-volume liquid flow rates. In addition, it replaces manual calculations, significantly reducing human error and enhancing both measurement accuracy and system robustness. This technological advancement not only improves reliability but also streamlines the detection process in dynamic industrial environments.

Energetic liquids typically exhibit transparent crystalline particulate properties, leading to reflection phenomena [5]. When using visual detection, these reflections pose higher demands on the accuracy of the detection system. The model must maintain high precision under complex lighting conditions. Additionally, height differences between the two pointers can cause depth-of-field issues during imaging, leading to variations in pointer imaging and challenging the model’s learning ability and parameter optimization. Moreover, since the system determines liquid viscosity based on the detection timing of pointers, the model must ensure rapid inference speed to meet response requirements.

Traditional liquid detection methods primarily rely on physical contact measurements [6,7,8]. However, as the demand for precision and real-time performance increases, the limitations of these traditional methods become more apparent. In recent years, visual detection technology has emerged as a key research focus due to its non-contact and high-precision characteristics. The evolution of liquid visual detection has progressed from simple image processing techniques to complex systems incorporating deep learning models [9]. Early visual detection systems primarily relied on basic algorithms such as image contrast and edge detection [10]. However, recent technological advancements have enabled these systems to handle more complex liquid properties and operating environments. The integration of AI and computer vision has significantly enhanced detection accuracy and efficiency [11], offering broad applications in industrial automation and environmental monitoring.

Future research in liquid visual detection should focus on integrating advanced deep learning models and algorithms to enhance detection accuracy, efficiency, and applicability [12,13,14]. Wang et al. explored a Coriolis flowmeter model for gas–liquid two-phase flow, utilizing Support Vector Machine (SVM) and artificial neural network (ANN) for flow prediction. By integrating ANN, SVM, and Genetic Programming algorithms, the model’s generalization capability and prediction accuracy are enhanced, but it may lead to higher computational costs and take more training time [15]. Researchers have used wavelet signature extraction and artificial neural network methods to improve the performance of gas–liquid two-phase flowmeters in the oil and gas and petrochemical industries [16]. Drikakis reviewed machine learning applications in fluid dynamics, discussing challenges in liquid viscosity detection and potential future directions [17]. Zepel introduced a general computer vision system for monitoring liquid levels in various chemical experiments [18], while Tim et al. discussed a machine vision system using a single camera for automatic liquid level detection. This method utilizes only a single camera for liquid-level detection to reduce hardware costs. It also features a high degree of automation and strong adaptability. However, the algorithm’s complexity is high, and the method is affected by the transparency and reflectivity of the liquid [19]. Dejband’s team proposed a deep neural network (DNN) model for accurate water level identification, which offers high accuracy but may require a large amount of training data [20]. Lee et al. applied a recurrent neural network–long short-term memory (RNN-LSTM) model to real leakage data for feature extraction and classification, which provided excellent performance in time-series data but was computationally intensive [21]. He et al. used a genetic algorithm–convolutional neural network (GA-CNN) model for the real-time prediction of liquid level fluctuations in continuous casting molds, which allows for effective optimization but may be complex to implement [22]. Lin et al. present an enhanced YOLO algorithm for detecting floating debris in waterways. The improvements focus on optimizing detection speed and accuracy. The method excels in real-time performance and debris classification but faces challenges in dealing with overlapping objects and complex backgrounds [23]. The summary and analysis of the relevant research content mentioned above are presented in Table 1.

This study aims to integrate computer vision technology and deep learning models for liquid flow viscosity detection, proposing the DBN-AGS-FLSS model for liquid surface pointer detection through adaptive parameter optimization. Experimental analysis shows that this model effectively overcomes the challenges posed by varying detection environments, significantly improving detection efficiency and accuracy. The main contributions are as follows:Enhanced image contrast and detection accuracy using bilateral filtering and adaptive contrast enhancement (ACE), maintaining edge contours and eliminating noise under varying lighting conditions.Achieved precise edge detection of liquid surface images using Canny operator mathematical morphology filtering and subpixel subdivision algorithms, enabling real-time feedback-based adaptive parameter optimization for continuous accuracy and robustness improvements.Proposed the DBN-AGS-FLSS deep learning model for real-time liquid surface pointer detection, incorporating reverse judgment feedback into the learning process, enhancing detection performance and robustness through adaptive optimization.

The detection model proposed in this paper firstly obtains real-time images of the liquid surface through a network camera and then performs image filtering and enhancement. Then, the liquid surface pointer is effectively detected by the DBN-AGS-FLSS deep learning model. A reverse judgment algorithm based on computer vision geometric parameter measurement is used to verify the detection results and feed the reverse judgment results into the deep learning model for parameter optimization. Finally, the flow rate and viscosity of the liquid are calculated by recording the time of the upper and lower pointers on the liquid surface and calculating the time difference. The specific workflow of this study is illustrated in Figure 1.

## 2. Materials and Proposed Method

The detection model proposed in this paper calculates liquid flow and viscosity by recording the timing of the upper and lower pointers appearing on the liquid surface and computing the time difference. The process begins with filtering and enhancing real-time liquid video images collected by the camera. The DBN-AGS-FLSS deep learning model is then introduced to efficiently detect the liquid surface pointers, complemented by a reverse judgment algorithm based on computer vision geometric parameter measurements. The reverse judgment results are fed back into the deep learning model for parameter optimization, effectively enhancing the model’s accuracy and robustness. The flowchart of the detection algorithm proposed in this study is shown in Figure 2.

### 2.1. Real-Time Image Enhancement Algorithm Based on Bilateral Filtering and ACE

The liquid surface video images captured by the camera are susceptible to hardware influences, environmental interference, and noise introduced by the surface texture and inherent properties of the liquid. Thus, it is essential to preprocess these images through filtering and enhancement. Given the small size of the pointer tip, its imaging area and pixel count in the image are minimal. In low-light conditions, the contrast between the liquid surface and the pointer tip is also relatively low. To meet detection requirements, it is necessary to optimize the camera’s imaging environment and use image processing techniques to eliminate interference and enhance contrast. This ensures that the system can accurately and promptly detect and record the moment the pointer appears on the liquid surface.

#### 2.1.1. Bilateral Filtering

Traditional filtering algorithms only consider the spatial distance between pixels, resulting in some degree of smoothing and blurring. Given the small imaging area and pixel count, using traditional methods may damage the target region, leading to detection failure. To address this, we employ a bilateral filtering algorithm that not only suppresses noise but also preserves edges, ensuring the integrity of the target area during the filtering process [24].
(1)$$g\left(x,y\right)=f\left(x,y\right)+n\left(x,y\right)$$

The zero-mean additive Gaussian noise model is shown in Equation (1), f represents the noise-free image, n is the noise following a zero-mean Gaussian distribution, and g is the noisy image with pixel value $g\left(x,y\right)$ at position $\left(x,y\right)$.

To suppress the noise n in image g and reconstruct the denoised image f, bilateral filtering is employed. This method, similar to Gaussian filtering [25], uses local weighting.
(2)$$f^{\prime }\left(x,y\right)=\frac{\sum_{\left(i,j\right)\in S_{\left(x,y\right)}}w\left(i,j\right)g\left(i,j\right)}{\sum_{\left(i,j\right)\in S_{\left(x,y\right)}}w\left(i,j\right)}$$

In Equation (2), $S_{\left(x,y\right)}$ represents the local neighborhood centered at $\left(x,y\right)$ of size (2N+1)×(2N+1). The weight $w\left(i,j\right)$ is composed of two factors, as shown in Equations (3) and (4).
(3)$$w_{s}\left(i,j\right)=e^{-\left(\frac{{\left|i-x\right|}^{2}+{\left|j-y\right|}^{2}}{2\delta_{s}^{2}}\right)}$$
(4)$$w_{r}\left(i,j\right)=e^{-\left(\frac{{\left|g\left(i,j\right)-g\left(x,y\right)\right|}^{2}}{2\delta_{r}^{2}}\right)}$$

It is evident that the weighting coefficient $w\left(i,j\right)$ of the bilateral filter is the product of two factors: the geometric proximity factor $w_{s}$ and the luminance similarity factor $w_{r}$. The value of $w_{r}$ is influenced by the pixel value differences within the neighborhood. When these differences are small, indicating a relatively flat image, $w_{r}$ approaches 1, making $w_{s}$ dominant, akin to Gaussian blur. As differences increase, $w_{r}$ diminishes, preserving edge information. The neighborhood radius N controls the extent of the smoothing effect—when N decreases, the smoothing effect also weakens. The standard deviation $\delta_{s}$ determines the rate of decay for $w_{s}$, while $\delta_{r}$ controls the rate of decay for $w_{r}$ [26].

#### 2.1.2. ACE for Liquid Level Pointer Image

ACE addresses the issue of low contrast between the liquid surface and the pointer tip in dim lighting conditions. After bilateral filtering, image enhancement is necessary to ensure clarity and ease of feature extraction. To mitigate interference from low-frequency backgrounds, a “local” enhancement method is employed for better results. The ACE algorithm processes the image in two stages: first, the original image is smoothed to obtain the low-frequency component; second, the high-frequency component is derived by subtracting the low-frequency part from the original image. The high-frequency component is then enhanced, and the two components are recombined to produce the final output image.

Step 1: Assume $x\left(i,j\right)$ is the grayscale value of a point in the image. The low-frequency component is obtained by calculating the local average, using a template of size (2N+1)×(2N+1), as expressed in Equation (5). The local variance is calculated using Equation (6).
(5)$$m_{x}\left(i,j\right)=\frac{1}{{\left(2n+1\right)}^{2}}\sum_{k=i-n}^{i+n}\sum_{l=j-n}^{j+n}x\left(k,l\right)$$
(6)$$\delta_{x}\left(i,j\right)=\frac{1}{{\left(2n+1\right)}^{2}}\sum_{k=i-n}^{i+n}\sum_{l=j-n}^{j+n}{\left[x\left(k,l\right)-m_{x}\left(i,j\right)\right]}^{2}$$

Step 2: Enhance the high-frequency component. The gain is determined by the standard deviation, with the mean $m_{x}$ approximating the background. Thus, $x\left(i,j\right)-m_{x}\left(i,j\right)$ represents the high-frequency details, as expressed in Equation (7) [27].
(7)$$f\left(i,j\right)=m_{x}\left(i,j\right)+\frac{D}{\delta_{x}\left(i,j\right)}\left[x\left(i,j\right)-m_{x}\left(i,j\right)\right]$$

The effects after bilateral filtering and ACE processing are shown in Figure 3.

### 2.2. Real-Time Detection of Liquid Surface Pointers Based on the DBN-AGS-FLSS Integrated Deep Learning Model

We propose a comprehensive deep learning model, DBN-AGS-FLSS, integrating DBN, AGS, and FLSS for the real-time detection of liquid surface pointers. The DBN performs unsupervised learning on preprocessed liquid surface images, capturing advanced features such as contrast and geometric shapes. An adaptive loss function embedded with reverse judgment feedback enhances the global performance of the Least-Squares Support Vector Machine (LSSVM). Additionally, the feedback from reverse judgment is incorporated into the genetic algorithm’s fitness function, combining feature extraction effectiveness $\theta_{ex}$, detection accuracy $\theta_{ac}$, and the evaluation of detection results relative to actual measurements $\theta_{con}$. This mechanism allows the model to dynamically adjust classification boundaries in real time based on feedback, significantly improving detection accuracy, robustness, and adaptive learning in dynamic environments.

#### 2.2.1. DBN Feature Extraction

In the detection of liquid surface pointers, timeliness and accuracy are crucial. Therefore, we selected DBN as the feature extraction method, which is capable of processing complex image data quickly and efficiently [28]. DBN is a type of deep neural network composed of hidden units and stacked Restricted Boltzmann Machines (RBMs). These RBMs serve as the building blocks of the DBN, with each layer learning different feature representations of the input data, enabling the network to capture complex patterns. By stacking multiple RBM layers, DBN learns hierarchical representations, achieving effective feature extraction. Unlike traditional deep learning models like the convolutional neural network (CNN) and recurrent neural network (RNN), DBN includes an unsupervised pre-training phase, helping to mitigate the vanishing gradient problem [29]. Additionally, DBN does not rely on labeled data during initial training, making it effective for handling unlabeled data. Furthermore, DBN possesses generative capabilities, allowing it to generate new data samples by learning the underlying structure and relationships within the data, offering deeper insights compared to purely discriminative models like CNN and RNN.

Figure 4 illustrates the architecture of the DBN, where each RBM consists of an input layer e and a hidden layer h. The neurons in each layer operate independently, with the input layer represented as $e=\{e_{1},e_{2},...,e_{m}\}$ and the hidden layer as $h=\{h_{1},h_{2},...h_{n}\}$. Through unsupervised training, the RBM learns complex hierarchical representations of the data [30]. These representations are then utilized in subsequent layers of the DBN for tasks such as classification or generation. The DBN excels in capturing intricate patterns within data, making it highly effective for handling high-dimensional data and complex tasks.

In DBN, multiple RBM layers are typically included. While the addition of these layers can enhance the model’s representational capacity, it also significantly increases computational complexity. This complexity not only prolongs training time and resource consumption but may also lead to overfitting. In the detection of liquid surface pointers, real-time performance and efficiency are paramount. Reducing the number of layers appropriately has a positive impact on improving the real-time performance and resource utilization of the detection system. Therefore, this study optimizes the DBN network architecture by comparing the performance of models with different numbers of RBM layers in liquid surface pointer detection. This optimization ensures that detection accuracy is maintained while improving detection efficiency and resource utilization.

The DBN method is employed to extract liquid surface features, involving both pre-training and fine-tuning stages. In the pre-training stage, various liquid surface data are passed through the first RBM layer of the DBN via unsupervised learning. The output from this trained layer is fed into the next, initiating a layer-by-layer learning process that continues until the final layer, generating features of the liquid surface pointer. In the fine-tuning stage, the results are compared with labeled data. Error backpropagation is used to iteratively adjust DBN parameters and enhance performance.

After the initial processing of the liquid surface images, a DBN model is deployed, as detailed in Table 2, including the output shape of each layer, activation functions used, regularization methods, and dropout rates. This model is designed to extract meaningful features from the preprocessed data, with each layer capturing increasingly abstract features. After training is complete, the DBN model can be copied up to the final dropout layer, excluding the final classification layer, to obtain a feature extractor model. This extractor is then used for predictions, effectively condensing high-dimensional data into a lower-dimensional feature space. Subsequently, the extracted features are classified using FLSS, with AGS employed for hyperparameter optimization.

#### 2.2.2. FLSS Real-Time Classification

The LSSVM is an improvement of the traditional SVM algorithm [31]. By introducing a least-squares loss function, it transforms the problem into a set of linear equations, thereby simplifying the optimization process. This approach enhances computational efficiency, making it particularly suitable for large-scale datasets.

After feature extraction, LSSVM is employed as the top-level classifier model in DBN for real-time classification [32]. In standard SVM, the loss function used is Hinge Loss [33], as expressed in Equation (8).
(8)$$L\left(\omega ,b\right)=\frac{1}{2}{\left\|\omega \right\|}^{2}+C\sum_{i=1}^{m}max(0,\;1-y_{i}(\omega^{T}x_{i}+b))$$

In LSSVM, the loss function is simplified into a quadratic optimization problem. The objective function is given by Equation (9), with the corresponding constraint conditions outlined in Equation (10).
(9)$$L\left(\omega ,r,v\right)=\frac{1}{2}{\left\|\omega \right\|}^{2}+\frac{\gamma }{2}\sum_{i=1}^{m}{v_{i}}^{2}$$
(10)$$y_{i}(\omega^{T}\varphi (x_{i})+r)=1-v_{i}$$

The first term $\frac{1}{2}{\left\|\omega \right\|}^{2}$ is a regularization term used to control the model’s complexity. The second term $\frac{\gamma }{2}\sum_{i=1}^{m}{v_{i}}^{2}$ differs from standard SVM. LSSVM uses a quadratic loss to penalize samples that violate constraints. The parameter γ is the penalty coefficient, controlling the model’s tolerance to errors. The term $\varphi (x_{i})$ represents the result of mapping the input into a high-dimensional feature space via a kernel function, and $v_{i}$ is a slack variable allowing some samples to not strictly satisfy boundary conditions. The classification decision function for LSSVM is shown in Equation (11), where x represents the feature vector extracted by DBN, and $K\left(x,x_{a}\right)$ is the kernel function. In this study, the Radial Basis Function (RBF) is selected as the kernel function, as represented in Equation (12). This indicates minimizing the error during training by mapping data into a high-dimensional space using the RBF kernel and performing classification. Through this approach, LSSVM can quickly and effectively classify the pointer positions in liquid surface images.
(11)$$z\left(x\right)=sign\left[\sum_{i=1}^{m}\alpha_{i}y_{i}K\left(x,x_{a}\right)+r\right]$$
(12)$$K\left(x_{a},x_{b}\right)=e^{-2{\left\|x_{a}-x_{b}\right\|}^{2}/\sigma^{2}}$$

In the context of liquid surface detection in this study, the model must handle complex physical environments (such as reflections and varying liquid densities), where traditional classification loss alone may be insufficient to ensure reliability. To address this, we introduce reverse judgment based on computer vision geometric parameter measurements and integrate its feedback into the loss function, proposing the FLSS method. FLSS dynamically adjusts the weights and classification boundaries of LSSVM, enhancing the classifier’s overall performance and increasing the model’s credibility in real applications.
(13)$${Loss}_{v}=\sum_{i=1}^{m}{\left({S^{\prime }}_{i}-S_{i}\right)}^{2}$$

In Equation (13), ${Loss}_{v}$ represents the difference between the model’s predicted values and the actual values determined by reverse judgment. Here, $S^{'}$ denotes the predicted pointer position by the model, while S is the position calculated based on computer vision geometric parameters (the detailed calculation will be explained in Section 3.3). During training, ${Loss}_{v}$ is used as part of the loss function to adjust the model’s weights and parameters. The loss function is expressed in Equation (14).
(14)$${Loss}_{total}=\lambda_{1}{\times (\frac{1}{2}{\left\|\omega \right\|}^{2}+\frac{\gamma }{2}\sum_{i=1}^{m}{v_{i}}^{2})+\lambda }_{2}\times \sum_{i=1}^{m}{\left({S^{\prime }}_{i}-S_{i}\right)}^{2}$$

In Equation (14), the first part represents the traditional classification loss of LSSVM, which measures the model’s accuracy in classification tasks. The second part reflects the difference between the model’s predicted values and the actual values determined by reverse judgment. By adjusting the weight coefficients $\lambda_{1}$ and $\lambda_{2}$, the impact of each loss component on model training can be controlled. This loss function design enables FLSS to continuously refine its classification decisions during training, minimizing errors and ensuring high accuracy and robustness under varying detection conditions.

#### 2.2.3. AGS Hyperparameter Optimization

A genetic algorithm (GA) is an optimization algorithm based on natural selection and genetic mechanisms, designed to find the optimal solution to a problem [34]. GA begins by initializing a population, where each individual represents a combination of hyperparameters for DBN and FLSS. During each generation, GA evaluates each individual in the population using a fitness function and generates new parameter combinations through selection, crossover, and mutation. Over multiple generations, GA converges on an optimal set of parameters.

AGS is an optimization mechanism that combines reverse judgment feedback with GA. Experimental analysis shows that this mechanism not only improves hyperparameter optimization efficiency but also enhances detection accuracy and robustness through dynamic model adjustments based on feedback. During evaluation, a comprehensive fitness function was designed, where θ represents the set of model hyperparameters. This function considers not only the feature extraction performance $\theta_{ex}$ and detection accuracy $\theta_{ac}$ but also incorporates a score $\theta_{con}$ comparing detection results with actual measurements.
(15)$$\theta_{ac}=\frac{\theta_{TP}+\theta_{TN}}{\theta_{TP}+\theta_{TN}+\theta_{FP}+\theta_{FN}}$$

In Equation (15), $\theta_{TP},\theta_{TN},\theta_{FP},\theta_{FN}$ represent the counts of true positives, true negatives, false positives, and false negatives, respectively.
(16)$$\theta_{ex}=1-\frac{1}{n}\;\sum_{i=1}^{n}\left|\left({X^{\prime }}_{i}-X_{i}\right)\right|$$

The feature extraction effectiveness can be defined by the reconstruction error of the DBN during the unsupervised learning phase, as shown in Equation (16). Here, ${X^{'}}_{i}$ represents the sample reconstructed by the DBN, $X_{i}$ is the original input sample, and n is the number of samples.
(17)$$\theta_{con}=1-\frac{1}{m}\;\sum_{i=1}^{m}\left|{{(S}^{\prime }}_{i}-S_{i})\right|$$

In Equation (17), ${S^{\prime }}_{i}$ represents the predicted value by the model, $S_{i}$ is the value measured by the computer vision reverse judgment, and m is the number of samples.
(18)$$\theta =\alpha \times \theta_{ac}+\beta \times \theta_{ex}+\gamma \times \theta_{con}$$

The improved fitness function is expressed in Equation (18), where α,β,γ are weight coefficients used to balance the influence of each component on the fitness function. During the AGS optimization process, we aim to find a hyperparameter combination $\left(\theta \right)$ that maximizes detection accuracy, feature extraction effectiveness, and consistency with reverse judgment. This optimization not only enhances the feature extraction quality of DBN but also helps FLSS achieve more precise classification, thereby minimizing the overall system loss function. Through this fitness function design, AGS effectively selects parameter combinations that exhibit high detection accuracy and strong consistency with reverse judgment results, significantly improving the model’s overall performance. The detailed process of hyperparameter optimization based on AGS is illustrated in Figure 5.

### 2.3. Reverse Judgment Algorithm for Liquid Surface Pointer Detection Based on Computer Vision Geometric Parameter Measurement

In this study, the camera’s position relative to the container is fixed, creating a proportional relationship between the geometric parameters of the circular liquid surface area and the liquid height within the container as the surface lowers. Therefore, the geometric parameters of the circular liquid surface area can be detected and measured in real time using computer vision-based geometric parameter measurement techniques. These parameters are then used for reverse judgment to verify the accuracy of the detected pointer position, ensuring precision throughout the detection process.

The core of reverse judgment in pointer detection lies in geometric parameter measurement based on computer vision. By measuring the radius of the circular region using computer vision and calculating parameters such as area and perimeter, the accuracy of pointer detection can be assessed by comparing these calculated results with the visual detection outcomes. This process includes three key components: coarse localization using Canny-operator-based edge detection, morphological filtering, and precise edge detection using subpixel subdivision algorithms.

#### 2.3.1. Image Edge Detection and Localization

In this study, the Canny algorithm with “dual-threshold” characteristics is used to effectively identify edge contours [35]. The implementation process involves smoothing the image with a filter, calculating the gradient magnitude and direction, performing non-maximum suppression on the gradient magnitude, and finally applying thresholding and edge linking to identify the edges.

The Canny algorithm segments the image using dual thresholds $T_{h}$ and $T_{l}$. For a point (x,y) with a gradient magnitude P(x,y), if P(x,y) > $T_{h}$, the point is confirmed as an edge point. If P(x,y) < $T_{l}$, it is excluded from being an edge point. If $T_{l}$ < P(x,y) < $T_{h}$, it is considered a potential edge point. Further analysis is performed based on the characteristics of neighboring pixels to determine if it should remain classified as an edge point, as shown in Figure 6.

The setting of the high threshold $T_{h}$ is critical. A high $T_{h}$ reduces noise in the edge detection image but may result in incomplete edges. Conversely, a lower $T_{h}$ produces more complete contours but introduces more noise [36]. As shown in Figure 7, the extracted circular region edge effects are depicted with high thresholds of 0.13 and 0.02, respectively. For edge completeness, 0.13 was selected during verification, and complete edge contours were obtained after subsequent morphological filtering.

The liquid surface video images are coarsely localized using the Canny algorithm, resulting in a binary image of the circular liquid surface area. Edge detection may produce suboptimal outputs, including noise and unwanted connections or gaps between contours. Noise typically appears as dots, streaks, or small holes [37]. Morphological operations are employed to remove noise and maintain complete edge connections, with structural elements playing a key role. Selecting appropriate shapes and sizes for these elements significantly enhances detection efficiency. Structural elements possess two particularly important properties in the area to be detected: one is extensibility, and the other is the effectiveness of the filling method. These properties are crucial for obtaining information about the target object [38].

Selecting the appropriate structural element for the erosion operation is important. The filtering effect of the structural element varies with its size and shape; smaller elements have weaker filtering capabilities. Experimental validation has shown that circular structural elements work best, so we create circular elements of suitable size for morphological operations. The original image is denoted as A, and the circular structural elements with radii of 2 and 1 are denoted as B. Equation (19) represents the dilation operation, which enlarges the target area and can fill small holes or eliminate minor noise. Equation (20) describes the erosion operation, which shrinks the target area to suppress small, unwanted objects. The opening operation process is described by Equation (21), and the closing operation by Equation (22). The final noise optimization of the edge detection results is achieved through the repeated application of Equations (19)–(22).
(19)$$F=A⊕B=\left\{x,y\left|{\left(B\right)}_{xy}\cap A\neq \emptyset \right.\right\}\;$$
(20)$$F_{1}=A\Theta B=\left\{x,y\left|{\left(B\right)}_{xy}\subseteq A\right.\right\}\;$$
(21)$$F_{2}=F\Theta B=\left(A⊕B\right)\Theta B$$
(22)$$F_{3}=F⊕B=\left(A\Theta B\right)⊕B$$

Traditional edge detection algorithms only achieve integer-level pixel accuracy, while subpixel subdivision algorithms can achieve decimal-level pixel precision. Subpixel subdivision is a software-based improvement, where achieving a coordinate precision of 0.1 pixels is equivalent to a tenfold increase in hardware resolution [39]. This method is not only cost-effective and fast but also significantly enhances measurement accuracy. The polynomial interpolation subpixel subdivision algorithm is an edge detection method that performs interpolation on target information to derive the optimal polynomial solution, which is then precisely applied across all data points [40]. Initially, the pixel is positioned at a one-pixel width location, and subsequent interpolation is used to determine the subpixel coordinate points. The edge detection result using the polynomial interpolation method is illustrated in Figure 8.

#### 2.3.2. Reverse Judgment for Liquid Surface Pointer Detection

After applying subpixel subdivision and other processing techniques, we use the least-squares method to obtain the coordinates and radius of the circular region, as the area of interest is circular.

Let $S_{1}^{\prime }$ and $S_{2}^{\prime }$ represent the areas of the circular liquid surface region detected by the model at the positions of the upper and lower pointers. The true areas of the circular liquid surface region are denoted as $S_{1}$ and $S_{2\;}$, as illustrated in Figure 9.

The radii of the circles are denoted as $R_{1}$ and $R_{2}$, while the heights of the upper and lower pointer end faces are $h_{1}$ and $h_{2}$, respectively. The conical container angle is θ (a constant). Assuming at a certain moment the liquid surface height is $h\in [h_{1}$, $h_{2}]$, and the circular liquid surface area is $S\in [S_{1}$, $S_{2}]$, then S can be calculated using Equation (23). Assuming $C=\sin\left(\frac{\theta }{2}\right)$, the liquid surface height can be determined using Equation (24).
(23)$$S=\pi R^{2}=\pi h^{2}*\sin^{2}\left(\frac{\theta }{2}\right)$$
(24)$$h=\sqrt{\frac{S}{\pi C}}$$

By substituting $h_{1}$ and $h_{2}$ into the equations, the corresponding values $S_{1}$ and $S_{2}$ can be obtained. When the upper or lower pointer appears on the liquid surface, the relationship between $S_{1}^{\prime }$ and $S_{2}^{\prime }$ with $S_{1}$ and $S_{2}$ should satisfy the conditions outlined in Equation (27). This ensures that the detected pointer is indeed the actual, true pointer.
(25)$$S_{1}=\pi h_{1}^{2}*\sin^{2}\left(\frac{\theta }{2}\right)$$
(26)$$S_{2}=\pi h_{2}^{2}*\sin^{2}\left(\frac{\theta }{2}\right)$$
(27)$$\left|S_{1}^{\prime }-S_{1}\right|\leq \varepsilon_{1},\;\left|S_{2}^{\prime }-S_{2}\right|\leq \varepsilon_{2}$$

In summary, when the circular area measured by computer vision closely matches the calculated actual circular area, the probability is highest that the pointer currently appearing on the liquid surface is the true pointer. The recorded time at this moment is the most accurate, thus achieving the reverse judgment process for pointer detection results. This judgment outcome will be fed back into the loss function and fitness function of the DBN-AGS-FLSS model discussed in Section 2.2.

#### 2.3.3. Liquid Flow and Viscosity Calculation

After detecting the time difference between the appearance of the pointers using the aforementioned method, viscosity for different energetic liquids can be calculated using a specialized flow measurement funnel. This funnel is an industry-standard measuring tool. Liquid flow and viscosity are determined using Formulas (28) and (29).
(28)$$Q=\frac{V}{t_{2}-t_{1}}$$
(29)$$\eta =K\frac{t_{2}-t_{1}}{V}$$
where Q represents the flow rate, V denotes the liquid volume, and $t_{1}$ and $t_{2}$ indicate the times when the liquid surface passes the upper and lower pointers, respectively. η stands for viscosity, while K is a constant related to the funnel’s geometry and the properties of the liquid.

## 3. Experiment Results

### 3.1. Experimental Platform

The test system, as shown in Figure 10, mainly consists of an imaging module, a graphics processing workstation, and cable interface accessories. The imaging module includes a camera, a ring parallel light source, and a lens. The camera, equipped with a lens, captures real-time video images of the liquid surface inside the container and transmits them to the graphics processing workstation. The ring light source provides supplementary lighting to enhance image contrast during video capture. The workstation is configured with an Intel Core i9-14900HX CPU, NVIDIA RTX 4060 GPU, and 64 GB 5200 MHz memory.

For the core imaging device, we used the MER2-U3 series digital camera from DaHeng Imaging’s Mercury II generation, along with an HN series 20MP 1″ fixed-focus lens. The key specifications are shown in Table 3.

The system testing process is illustrated in Figure 11. Upon receiving a command, the system captures images through the camera and performs image enhancement. It then conducts real-time detection of the liquid surface pointers and carries out reverse judgment using geometric parameters. After both the upper and lower pointers have been recorded, the system calculates the time difference between them, which is subsequently used to compute the liquid’s flow rate and viscosity.

### 3.2. Experimental Evaluation Criteria

In the liquid surface pointer detection phase, we proposed the DBN-AGS-FLSS integrated deep learning model. To demonstrate its effectiveness, we selected recall, precision, F1 score, and accuracy as evaluation metrics [41]. These metrics are critical for assessing the performance and detection accuracy of classification algorithms.
(30)$$Recall=\frac{\sum_{\alpha }^{A}n_{\alpha }\times \left(\frac{TP_{\alpha }}{TP_{\alpha }+FN_{\alpha }}\right)}{N}$$
(31)$$Precision=\frac{\sum_{\alpha }^{A}n_{\alpha }\times \left(\frac{TP_{\alpha }}{TP_{\alpha }+FP_{\alpha }}\right)}{N}$$
(32)$$F1=\frac{1}{N}\sum_{\alpha }^{A}n_{\alpha }\times 2\times \sum_{\alpha }^{A}\left(\frac{{Recall}_{\alpha }\times {Precision}_{\alpha }}{{Recall}_{\alpha }+{Precision}_{\alpha }}\right)$$
(33)$$Accuracy=\frac{\sum_{\alpha }^{A}{TP}_{\alpha }}{N}$$

In classification algorithms, N represents the total number of samples in the test set, while $TP_{\alpha },FP_{\alpha },FN_{\alpha }$ correspond to the classification results for class A, indicating “true positive”, “false positive”, and “false negative”, respectively. A “false positive” occurs when a sample is incorrectly classified as class A, while a “true positive” indicates a correct classification of a sample as class A. A “false negative” occurs when a sample that should be classified as class A is misclassified into another category. Here, A represents the total number of different classes in the dataset, and $n_{\alpha }$ represents the total number of samples in class A. The sum of $TP_{\alpha }$ and $FN_{\alpha }$ gives the total number of samples that truly belong to class A, while the sum of $FP_{\alpha }$ and $n_{\alpha }$ represents the number of samples incorrectly classified as class A. As the number of samples increases, the accuracy of the algorithm’s evaluation improves.

In addition to traditional performance metrics such as accuracy, precision, F1 score, and recall, we also introduced inference speed as a key metric for evaluating the model’s effectiveness in real-time detection. Inference speed is crucial for assessing a deep learning model’s performance in real-time applications. Since this method relies on calculating the time difference as the liquid surface passes between two pointers, real-time capability is essential to ensure accurate measurements. A faster inference speed reduces latency, thereby enhancing the precision of liquid flow and viscosity measurements.

### 3.3. Experimental Results and Analysis

#### 3.3.1. Performance Evaluation of RBM Layer Configuration

In selecting the number of RBM layers in DBN, we compared five different configurations, with performance comparison results shown in Table 4. As the number of layers increases, the model’s accuracy gradually improves; however, the inference time and memory usage also increase correspondingly.

The increase in the number of RBM layers leads to a significant increase in computational complexity, training time, and resource consumption. With the increase in the number of layers, although the feature extraction ability of the model is enhanced, the accuracy is improved, but the inference speed and memory consumption will increase simultaneously. To further reduce the training time, we use an unsupervised pre-training phase, which allows the model to converge quickly with less labeled data. This also effectively reduces the overall training time and improves resource efficiency. We believe that there is a threshold on the number of RBM layers; that is, when the number of layers exceeds a certain value, the inference time and resource occupancy increase disproportionately, despite the improved accuracy of the model, and may lead to overfitting.

Our results indicate that the four-layer RBM model achieves an optimal balance between maintaining high detection accuracy and minimizing inference time and resource consumption. The inference time of the four-layer RBM model is 1.1 milliseconds, the accuracy rate is 99.2%, and the memory usage is 150 MB. This makes it the optimal choice for liquid surface pointer detection, ensuring both high detection precision and enhanced efficiency in resource utilization.

#### 3.3.2. Comparative Analysis of Different Models

The initial values for the parameters in the loss function are set to $\lambda_{1}=0.6,\;\lambda_{2}=0.4$, while in the fitness function, parameters are α=β=0.35, γ=0.3. The tolerance parameters for the error between the detected liquid surface pointer results and actual physical measurements are $\varepsilon_{1}=\varepsilon_{2}=0.01$. We compared the proposed model with other deep learning models like DNN, GA-CNN, RNN-LSTM, and You Only Look Once version 10 (YOLOv10) to validate its advantages. The comparison was conducted using three different datasets with liquid surface pointer exposures of 0.2 mm, 0.4 mm, and 0.8 mm. The dataset for each exposure consists of 640 samples. During the model training process, 70% of the samples are selected for training, and the remaining 30% are used for testing. Considering the reliability and consistency of the results, each dataset was tested 30 times separately. This approach helps validate the robustness and generalizability of the proposed method. The performance comparison results are shown in Table 5 and Table 6.

The comparison was based on four performance parameters, with each model undergoing 30 experiments across three different datasets. The results consistently show that the DBN-AGS-FLSS model proposed in this paper outperformed the other four models in all experiments and across all performance metrics. The DBN-AGS-FLSS model achieved higher accuracy, precision, F1 score, and recall, with values of 99.37%, 99.36%, 99.16%, and 99.36%, respectively. These findings indicate that the proposed method significantly enhances the reliability and accuracy of the liquid surface pointer detection system. To more intuitively demonstrate the performance of each model in actual classification tasks, a detailed comparative analysis was conducted using the confusion matrix shown in Figure 12 [42]. The results produced by the model proposed in this paper are more accurate than those of the four reference models.

As shown in Figure 13, the proposed DBN-AGS-FLSS model outperforms the three comparison models in terms of inference speed, with an average inference time of only 1.5 ms per image. The inference time was tested under consistent environment and hardware conditions. By inputting a single frame of image into the model and performing multiple tests, we calculated the average inference time. Specifically, the process involves recording the time when the single image frame is fed into the network and then recording the time again when the inference result is generated. The difference between these two times represents the inference time.

This not only ensures real-time performance in high-precision detection but also further validates the practicality and reliability of this method for industrial applications.

#### 3.3.3. Ablation Experiments of Each Module of the Model

After conducting performance evaluations with other models, we performed ablation experiments to further validate the contribution of each component within the overall model. The ablation experiment involved systematically removing key components from the DBN-AGS-FLSS model—namely, DBN, AGS, and FLSS—to observe their impact on the final detection performance. This approach highlights the contribution of each component to the model’s overall performance. We designed three control groups: using CNN instead of DBN; using GA instead of AGS; and replacing FLSS with standard SVM. The results of the ablation experiments are presented in Table 7 and Figure 14.

#### 3.3.4. Influence of Feedback Mechanism on Detection Accuracy

Figure 15 illustrates the recorded data for detected values and feedback values. The trend in the chart demonstrates that the error between the detected values and feedback values decreases progressively, eventually stabilizing around 0.001. This clearly illustrates the effectiveness of the reverse judgment feedback mechanism in correcting model parameters.

### 3.4. Actual Detection Effect

The actual detection images of the liquid surface pointer and hardware system architecture are shown in Figure 16 and Figure 17.

## 4. Discussion

### 4.1. Model Generalization Testing and Discussion

Although the experiments demonstrated that the algorithm performs well when dealing with different energetic liquids as the background, the energetic liquids we mainly deal with include nitrate ester liquids, nitric acid-based liquids, liquid rocket propellants, and water gas, among others. It still lacks validation in more complex environments, such as extreme lighting conditions (too bright or too dark), densely packed objects, or other industrial detection scenarios.

To validate the system’s performance on larger datasets and varying industrial applications, we conducted additional simple zero-sample experiments. Specifically, we tested on the CPPD (Comprehensive Pharmaceutical Packages) and EP (Express Packages) datasets, which primarily consist of densely packed objects with varying lighting conditions and dense item detection. The entire datasets were used for training, while the system was directly tested under zero-sample production conditions to observe its accuracy. The dataset information is provided in Table 8 and Table 9.

Additionally, the energetic liquid surface detection system requires detection to be conducted under constant lighting conditions. However, in order to verify the generalization of our model and its performance in different environments, we carried out experimental tests in low-illumination environments. The simple test results for the above three zero-sample scenarios are shown in Figure 18. Although the overall performance is not as strong as those of models specifically designed for certain solutions, the detection accuracy under zero-sample production conditions reached over 95%. This indirectly demonstrates that the model possesses a certain degree of generalization capability.

### 4.2. Discussion on the Model Inference Speed in Large-Scale Complex Scenarios

Although our inference speed on the small-scale dataset in this paper reached 1.5 ms per frame, when considering scalability to larger datasets or more complex liquid environments, we lack sufficient validation and deeper research. We plan to enhance efficiency through parallel computing techniques in the future, such as distributed and pipeline inference, which are particularly beneficial for multi-camera or large-scale monitoring systems. And by optimizing data flow, we aim to maintain real-time performance despite increasing complexity. Additionally, in challenging liquid environments with varying reflectivity and viscosity, adaptive feature extraction algorithms will help simplify irrelevant features, ensuring inference speed remains unaffected.

### 4.3. Discussion on the Limitations of the Model and Future Works

Although the algorithm achieved good results in the current detection environment, there are still some potential shortcomings. The algorithm’s performance is heavily dependent on accurate preprocessing, such as bilateral filtering and ACE. Inadequate preprocessing could compromise detection results. Additionally, hardware setup variations, like camera positioning, may introduce depth perception errors that the model cannot fully correct. Currently, the system relies solely on visual input for detection and correction, so incorporating technologies like laser ranging or 3D cameras could enhance reliability and provide extra safety measures.

Future research should prioritize the optimization of existing detection equipment and methodologies by incorporating more advanced deep learning models and algorithms. This focus aims not only for higher detection accuracy but also for a broader application scope, which can be crucial in addressing various industrial challenges. Additionally, exploring novel techniques such as transfer learning and ensemble methods may enhance model performance in complex environments. The potential future improvements are as below:Advanced deep learning models: Future research could involve the incorporation of more advanced deep learning algorithms and architectures, such as transformer-based models, to improve the robustness of the model under challenging conditions like extreme lighting or reflection.Integration with real-time monitoring systems: The current model shows promise in real-time liquid monitoring, but further optimizations could ensure even faster inference times, making it more compatible with industrial applications where real-time decision-making is critical.Improving robustness in extreme conditions: There is potential for enhancing the model’s robustness by introducing additional feedback loops or hybrid algorithms that can dynamically adjust the model parameters during extreme conditions, such as variable lighting or reflective surfaces. This could help maintain high accuracy regardless of environmental changes.Incorporating laser ranging for the real-time detection of liquid surface height, combined with multi-sensor fusion, can significantly improve the model’s performance by providing more accurate corrections and enhancing overall system reliability. The upgraded hardware structure is shown in Figure 19.

## 5. Conclusions

This research successfully presents an innovative approach for detecting liquid flow and viscosity, essential for various industrial and environmental applications. By leveraging advanced computer vision and deep learning techniques, the proposed DBN-AGS-FLSS model demonstrates high precision in identifying liquid surface pointers in real time. Additionally, the implemented feedback mechanism for reverse judgment effectively fine-tunes the model parameters, resulting in enhanced accuracy and reliability. The model was tested on three datasets and compared with DNN, GA-CNN, RNN-LSTM, and YOLOv10. Through ablation experiments, we identified the optimal number of layers for the model, balancing system performance and efficiency. Iterative testing verified the effectiveness of the system’s adaptive parameter tuning mechanism, ultimately keeping the error within 0.1%. Based on the above research, we confirmed its effectiveness across diverse detection scenarios. Finally, to validate the generalization capability of the model, we conducted zero-sample experiments in other industrial scenarios and datasets. We also discussed potential future research directions and optimizations in this field for further exploration. This work provides a reference for future advancements in liquid detection technologies, emphasizing the importance of continuous improvement and interdisciplinary collaboration in this field.

## Figures and Tables

**Figure 1 sensors-24-06733-f001:**
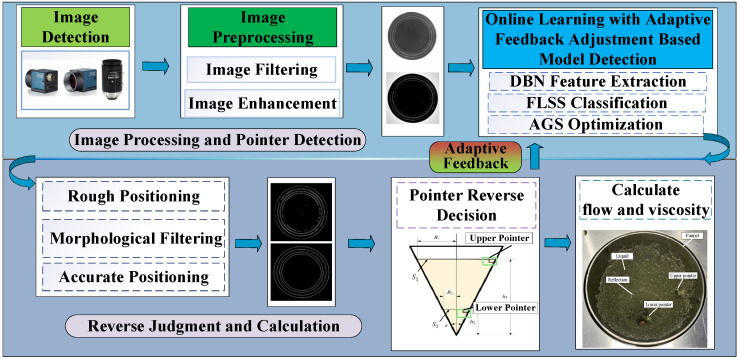
Workflow diagram of the entire study.

**Figure 2 sensors-24-06733-f002:**
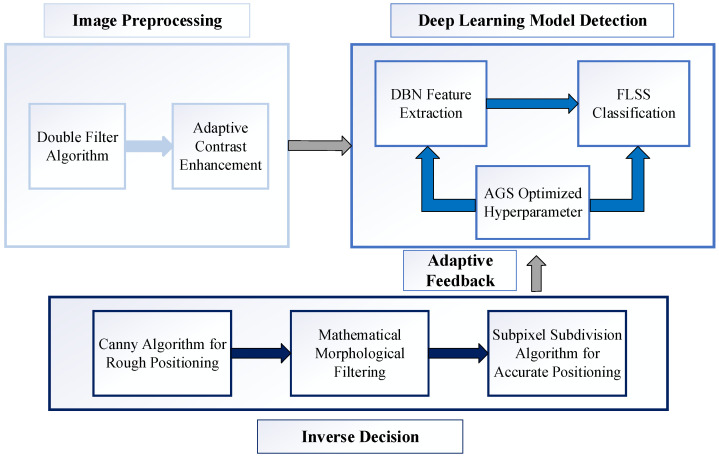
Algorithm flowchart.

**Figure 3 sensors-24-06733-f003:**
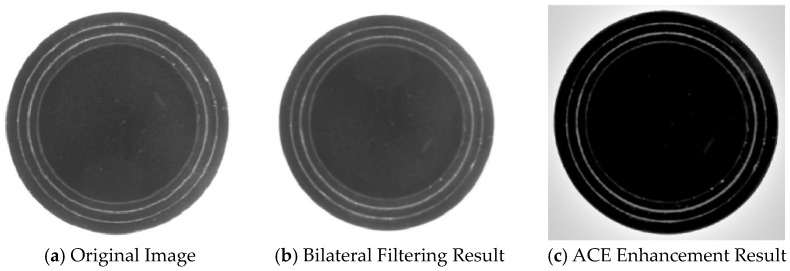
Image preprocessing results.

**Figure 4 sensors-24-06733-f004:**
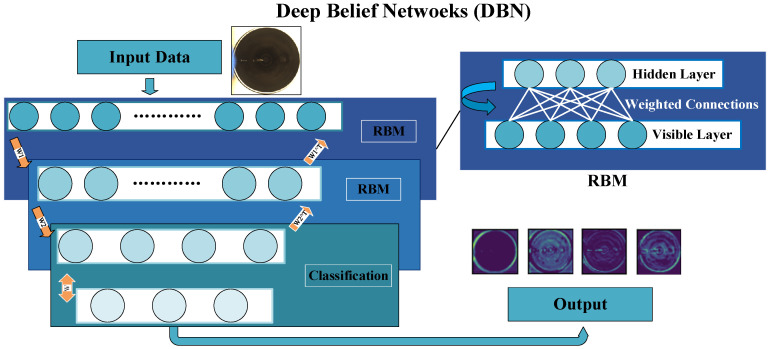
DBN architecture.

**Figure 5 sensors-24-06733-f005:**
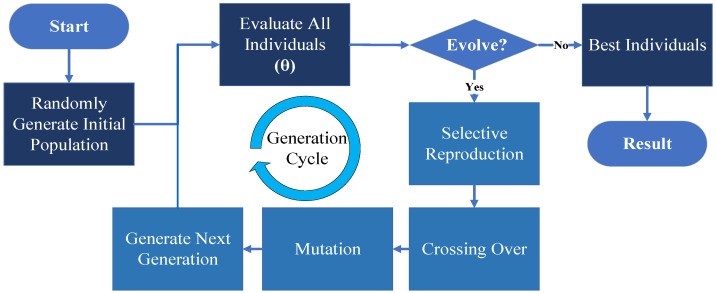
AGS-based hyperparameter optimization flowchart.

**Figure 6 sensors-24-06733-f006:**
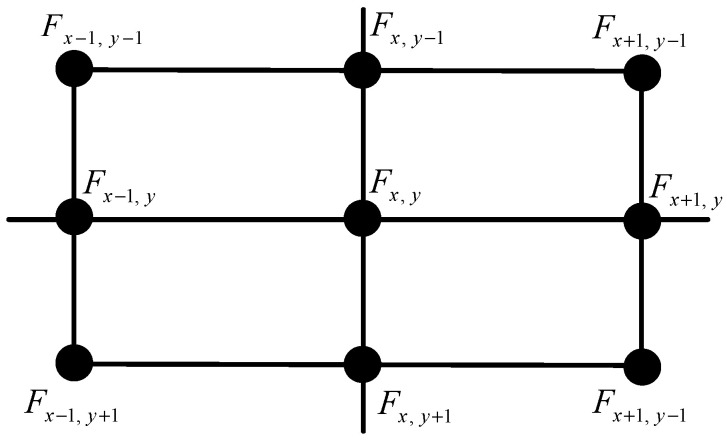
Neighborhood grayscale template.

**Figure 7 sensors-24-06733-f007:**
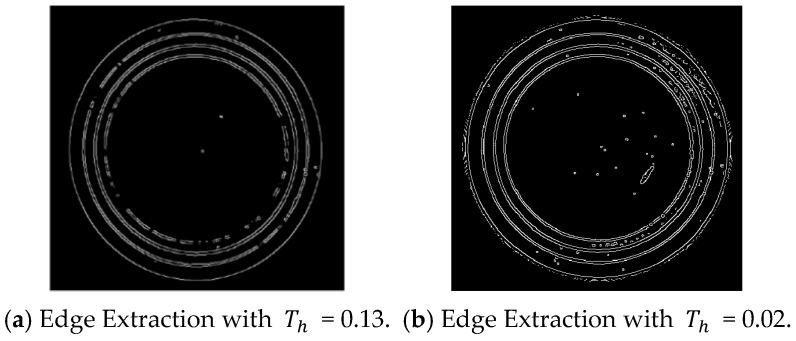
Edge contour map.

**Figure 8 sensors-24-06733-f008:**
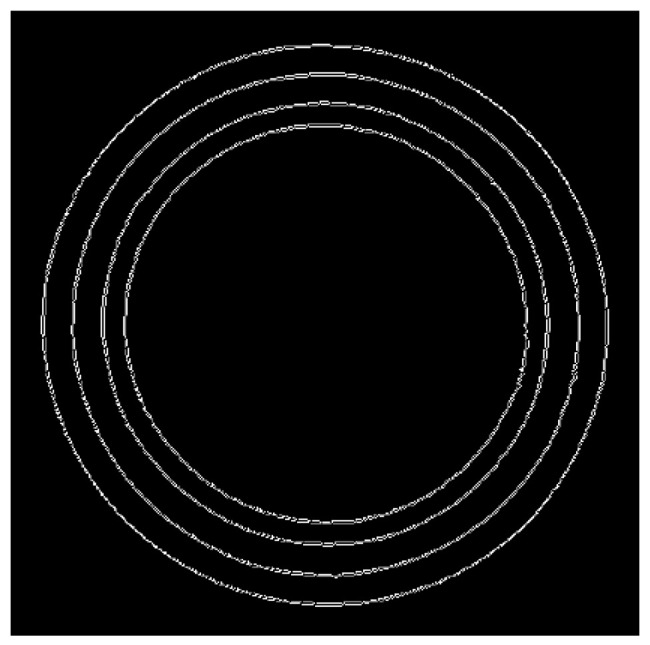
Edge detection result using polynomial interpolation method.

**Figure 9 sensors-24-06733-f009:**
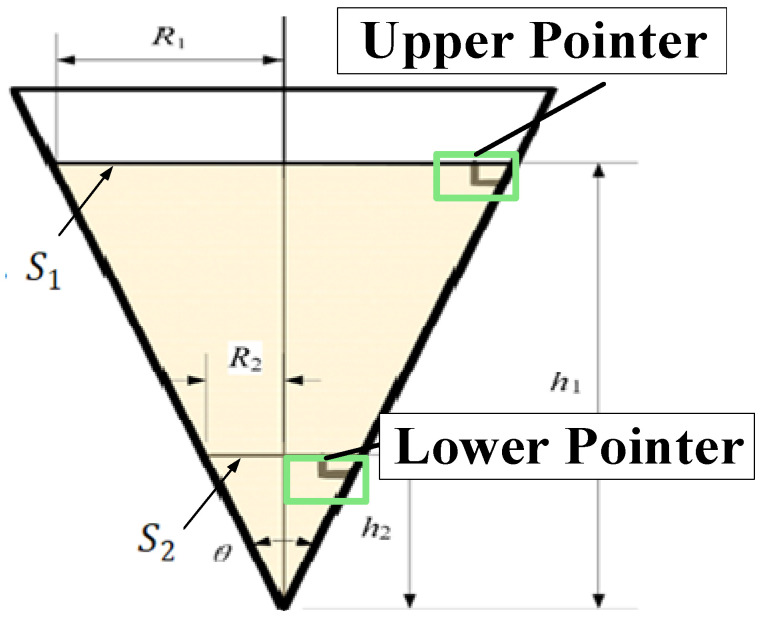
Liquid surface and upper-lower pointer.

**Figure 10 sensors-24-06733-f010:**
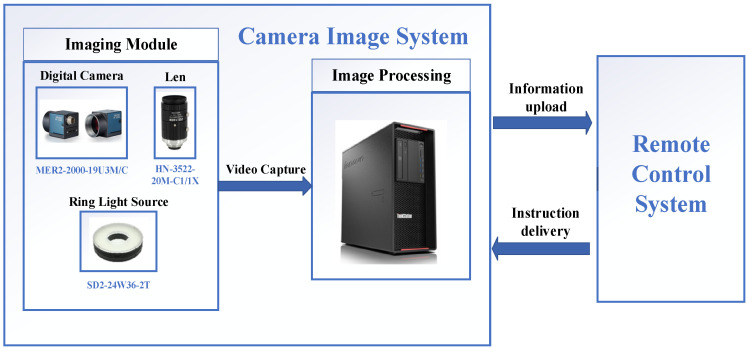
Test system configuration.

**Figure 11 sensors-24-06733-f011:**
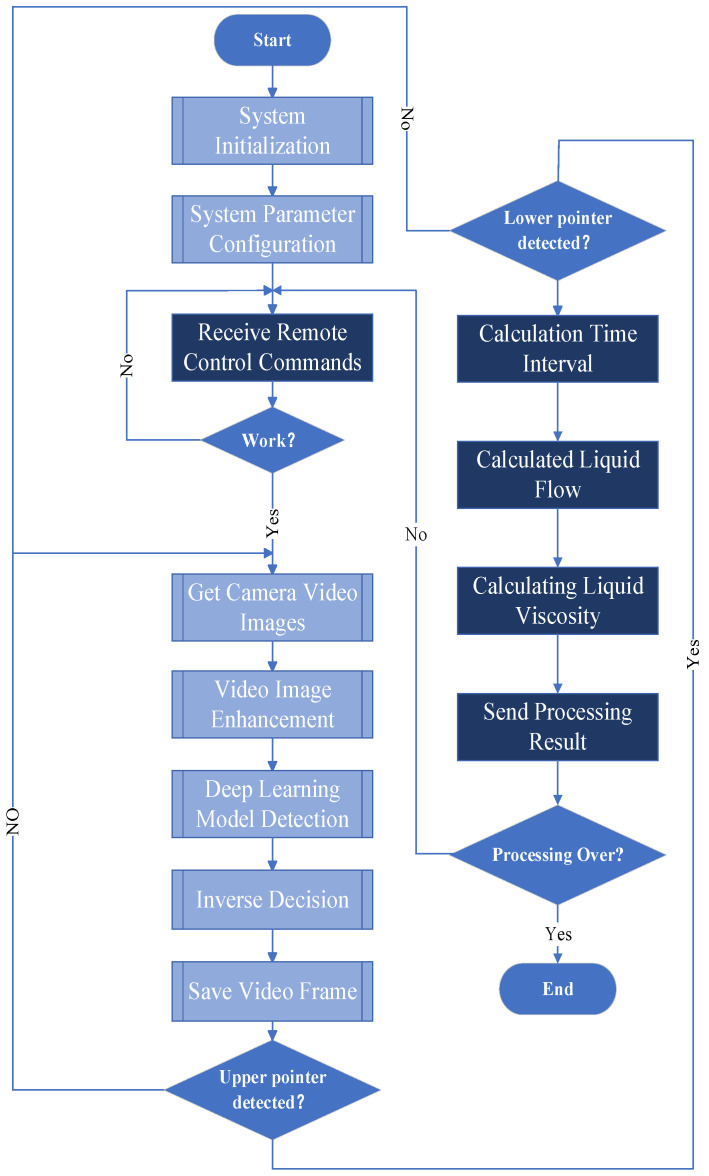
System testing flowchart.

**Figure 12 sensors-24-06733-f012:**
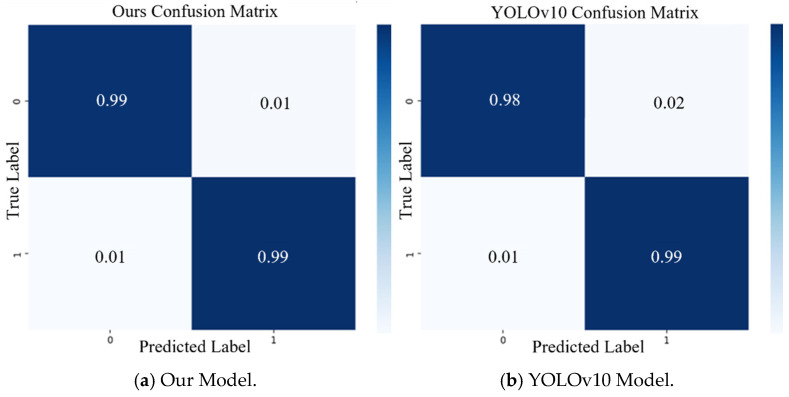
Confusion matrix comparison. Different shades of blue represent the proportion of correctly or incorrectly classified instances, with darker shades indicating higher accuracy (values close to 1) and lighter shades indicating lower accuracy.

**Figure 13 sensors-24-06733-f013:**
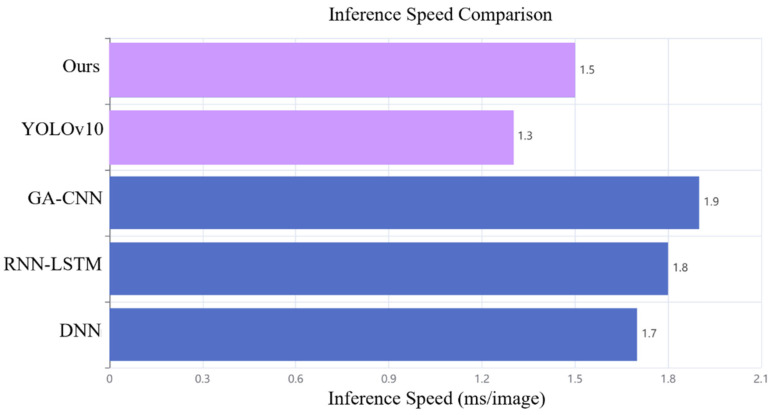
Inference speed comparison.

**Figure 14 sensors-24-06733-f014:**
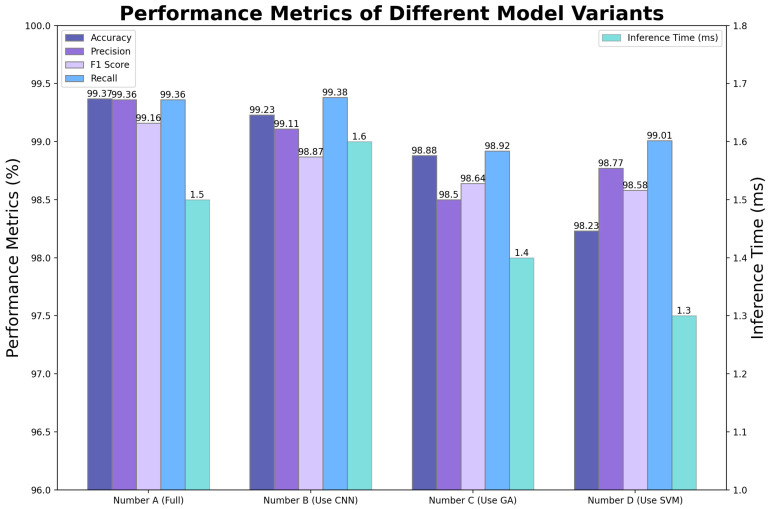
Ablation results with different models.

**Figure 15 sensors-24-06733-f015:**
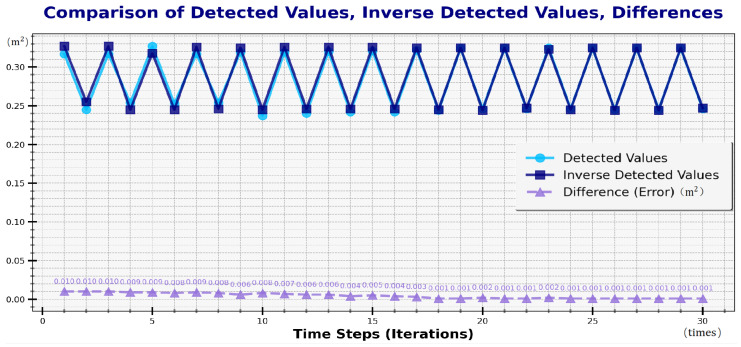
Detected values, feedback values, and error.

**Figure 16 sensors-24-06733-f016:**
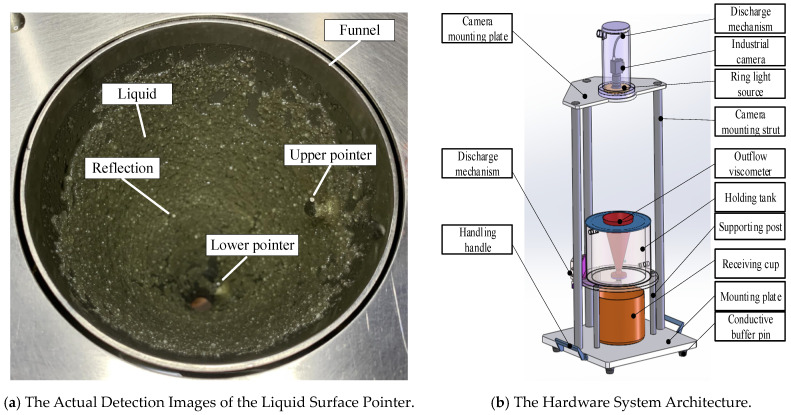
The actual detection images of the liquid surface pointer and hardware system architecture.

**Figure 17 sensors-24-06733-f017:**
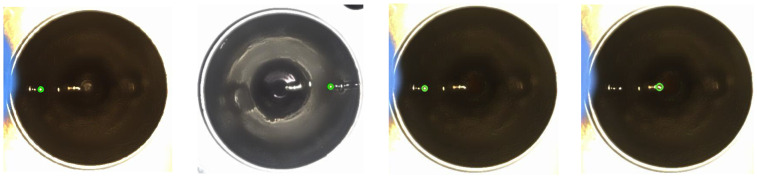
Actual detection results (detection samples).

**Figure 18 sensors-24-06733-f018:**
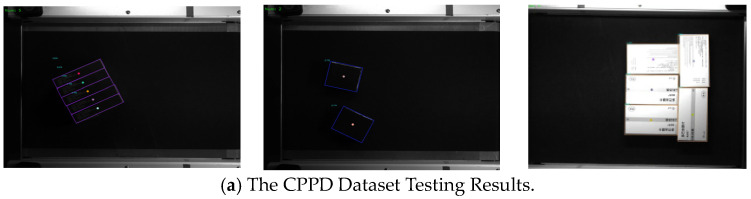
Detection performance in other datasets and scenarios.

**Figure 19 sensors-24-06733-f019:**
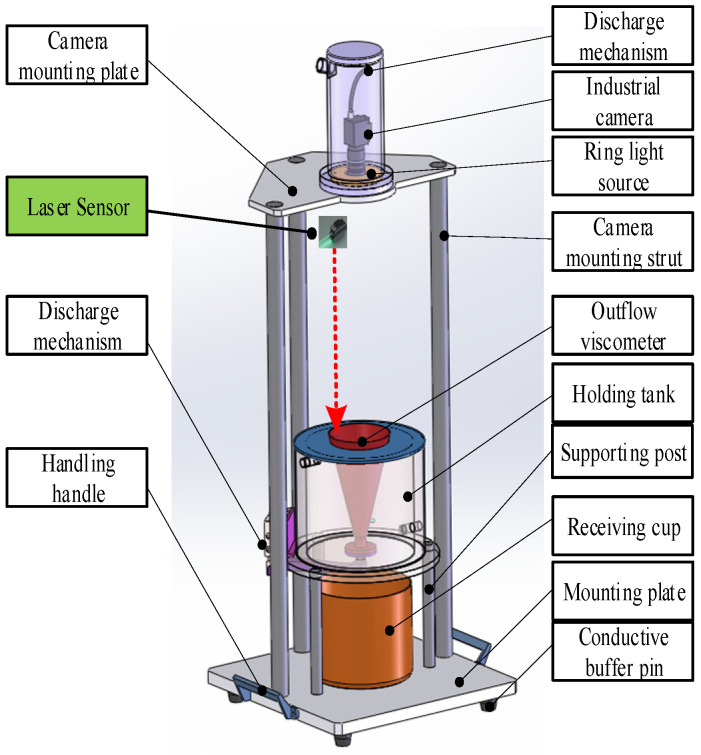
The upgraded hardware structure.

**Table 1 sensors-24-06733-t001:** Introduction to relevant models.

Model	Summary	Pros and Cons	Application
Artificial Neural Network (ANN)	Mimicking the way neurons in the brain work.	Non-linear mapping, generalization ability. Requires large datasets, lacks interpretability.	Image recognition, speech recognition, medical diagnosis.
Deep Neural Network (DNN)	Deep neural networks composed of multiple hidden layers.	High accuracy, scalability. Requires large amounts of data, overfitting.	Image recognition, speech recognition, natural language processing.
Recurrent Neural Network–Long Short-Term Memory (RNN-LSTM)	A recurrent neural network that processes time-series data.	Strong in handling time-series data, ability to retain information. Computationally expensive, slow training time.	Speech recognition, financial time-series forecasting, and leak detection in time-series data.
Genetic Algorithm–Convolutional Neural Network (GA-CNN)	Using GA to optimize CNN parameters, leading to improved performance on specific tasks.	Strong optimization capability, automated parameter selection. Complex implementation, slow convergence.	Real-time monitoring and optimization scenarios.
YOLO (You Only Look Once)	A real-time object detection algorithm that processes the entire image at once.	Real-time detection, high speed and accuracy. Struggles with small object detection, difficulty with overlapping objects.	Autonomous driving, video surveillance, traffic monitoring, and real-time object detection in various fields.

**Table 2 sensors-24-06733-t002:** DBN model parameter.

Layer	Output Shape	Activation	Regularization	Dropout
Dense	1024	ReLU	L2 (0.02)	None
Dropout	1024	-	-	0.4
Batch Normalization	1024	-	-	None
Dense	512	ReLU	L2 (0.02)	None
Dropout	512	-	-	0.4
Batch Normalization	512	-	-	None
Dense	256	ReLU	L2 (0.02)	None
Dropout	256	-	-	0.3
Batch Normalization	256	-	-	None
Dense	128	ReLU	L2 (0.02)	None
Dropout	128	-	-	0.3
Batch Normalization	128	-	-	None
Dense	64	ReLU	L2 (0.02)	None
Dropout	64	-	-	0.3
Batch Normalization	64	-	-	None
Dense	32	ReLU	L2 (0.02)	None
Dropout	32	-	-	0.3
Batch Normalization	32	-	-	None
Dense	16	ReLU	L2 (0.02)	None
Dropout	16	-	-	0.2
Batch Normalization	16	-	-	None
Dense	1	Sigmoid	None	None

**Table 3 sensors-24-06733-t003:** The main parameters of the camera for experiments.

No.	Device	Items		Parameter Value
1	Digital Camera (MER2-2000-19U3M/C)	Port		USB3.0
		Resolution rate		5496 (H) × 3672 (V)
		Frame rate		10 Hz19.6 fps @5496 × 3672
		Sensor		1″, Sony IMX183 Rolling shutter CMOS
		Pixel size		2.4 μm × 2.4 μm
		Spectrum		Black and white/Color
		Exposure time		12 μs~1 s
2	Lens (HN-3522-20M-C1/1X)	Focal length		35 mm
		Maximum aperture ratio		1:2.2
		Maximum imaging size		12.8 × 9.6 (Φ16)
		Aperture range		F2.2~F16.0
		Working distance		0.2 m~Inf.
		Field of view Angle (D × H × V°)	1″	25.8 × 20.8 × 15.7
			2/3″	17.9 × 14.4 × 10.8
			1/1.8″	14.7 × 11.8 × 8.9
		Rear focal length		9.7 mm

**Table 4 sensors-24-06733-t004:** RBM layer performance comparison.

No.	RBM Layers	Inference Time (ms)	Accuracy (%)	Memory Usage (MB)
A	2	0.7	98.7	90
B	3	0.9	98.9	120
C	4	1.1	99.2	150
D	5	1.5	99.3	190
E	6	2.1	99.5	240

**Table 5 sensors-24-06733-t005:** Performance evaluation.

Models	Accuracy (%)			Precision (%)			F1 Score (%)			Recall (%)		
Data	0.2	0.4	0.8	0.2	0.4	0.8	0.2	0.4	0.8	0.2	0.4	0.8
Ours	99.44	99.15	99.42	99.31	99.35	99.42	99.35	99.15	98.99	99.33	99.35	99.41
YOLOv10	98.61	98.57	98.66	99.22	99.12	98.69	98.67	98.79	99.23	99.25	99.31	98.66
GA-CNN	96.21	97.55	96.89	96.75	97.32	98.42	97.32	96.78	98.12	97.68	97.45	96.52
RNN-LSTM	95.78	97.35	98.25	97.63	98.44	97.24	96.77	95.98	97.24	98.36	97.15	96.87
DNN	94.31	93.76	94.56	94.61	93.27	95.89	96.67	95.27	94.72	93.57	94.32	95.68

**Table 6 sensors-24-06733-t006:** Average performance evaluation.

Models	Accuracy (%)	Precision (%)	F1 Score (%)	Recall (%)
Ours	99.37	99.36	99.16	99.36
YOLOv10	98.61	99.01	98.90	99.07
GA-CNN	96.88	97.50	97.41	97.22
RNN-LSTM	97.13	97.77	96.66	97.46
DNN	94.21	94.60	94.55	94.52

**Table 7 sensors-24-06733-t007:** Ablation results.

Number	Models	Accuracy (%)	Precision (%)	F1 Score (%)	Recall (%)	Inference Time (ms)
A	Full Model	99.37	99.36	99.16	99.36	1.5
B	Use CNN	99.23	99.11	98.87	99.38	1.6
C	Use GA	98.88	98.50	98.64	98.92	1.4
D	Use SVM	98.23	98.77	98.58	99.01	1.3

**Table 8 sensors-24-06733-t008:** Dataset introduction.

Datasets (By Ourselves)	Images	Objects	Describe
CPPD	16,848	151,632	Various types of medicine boxes made of various materials, covering all mainstream types of pharmacies, including challenges such as reflection caused by waterproof plastic film.
EP	25,127	60,393	A comprehensive sample covering all types of packages in the logistics and express delivery industry, with sizes ranging from 5 cm to 3 m and heights ranging from 0.5 mm to 1.2 m in various shapes.

**Table 9 sensors-24-06733-t009:** Dataset instance.

**Datasets**	**Examples**				
CPPD	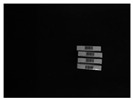	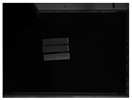	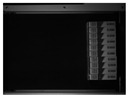	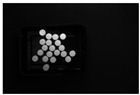	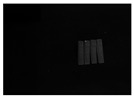
EP	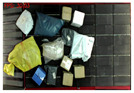	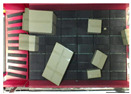	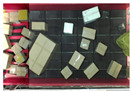	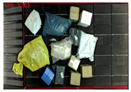	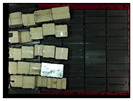

## Data Availability

The data analyzed or generated in this study are available from the corresponding authors upon reasonable request.

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
