# Peer review of "An Adaptive Parameter Optimization Deep Learning Model for Energetic Liquid Vision Recognition Based on Feedback Mechanism"

_sensors, 2024, doi:10.3390/s24206733_

Round 1

Reviewer 1 Report

Comments and Suggestions for Authors

In this study, based on the Deep Belief NetworksDBN)、Feed-back Least Squares SVM ClassifiersFLSS and Adaptive Genetic SelectorsAGS, a new deep learning model DBN-AGS-FLSS was proposed, which improves the image clarity and detection accuracy, and enhances the system accuracy and robustness. My comments are listed below:

1. Article construction needs to be tweaked. For example, related work section is suggested to be merged into the introduction section. To facilitate comprehension, it is essential to offer materials and methods section, results section, and discussion section. Given the inadequacy of your discussion material, I suggest combining the results section with the discussion section into the results and discussion section.

2. The table's format does not conform to standard criteria; please modify it.

3. Lines 33-36, Line40, Lines 43-45: references must be included to substantiate the sentences.

4. Line 36 ought to point out the limitations of conventional visual inspection while emphasizing the benefits of visual inspection technologies.  

5. Line 83: please confirm if it is appropriate to cite the 2001 publication that used neural networks for the study here.

6. Lines 81-94: recent research progress succinctly enumerate the used methodologies, omitting the outcomes as well as the advantages and disadvantages of the methods.

7. Section 3.1: covers the Bilateral Filtering Algorithm and the Adaptive Contrast Enhancement (ACE) Algorithm; Section 3.2 covers the DBN Algorithm, LSSVM Algorithm, and GA Optimization Algorithm. Please cite the reference if it is not the original algorithm.

8. Lines 210-216: this part is the basics of network training and should state which training set was used for pre-training and how many iterations were used for fine-tuning the parameters.

9. Lines 346-351: Statements defining properties should include references.

10. Lines 368-371: polynomial interpolation subpixel subdivision algorithm, should be added to the cited literature.

11. Section 3.3.2:  It could be abbreviated into a single-paragraph content and integrated into the other sections.

12. S1 and S2 mentioned in line 389 are not labelled in Fig 9, please label them clearly.

13. Lines 418-423, Fig 10: It is recommended that the figure includes images of real equipment combinations, and that camera models are specified in the text

14. Doesn't Deep Learning Model Detection connect to the image input in Fig 11, and why doesn't Get Camera Video Images connect to other modules?

15. Lines 464-469: this should be described here together with the significant data shown in Table 2.

16. Lines 476-481: 1. Does the total of 640 sample images include all three exposures, or does it represent 640 images for each exposure? Isn't this data quantity insufficient for a deep learning study such that the outcomes of several model performance evaluations mostly focus on 90% or higher? 2. Is the allocation of the training and test sets consistently 30% for the test set and 70% randomly designated for the training set? Is there a problem if the training set including the test set? 3. When testing different models, should the model be tested 30 times with the same model parameters but different test sets, or should the training and test sets be split each time to train 30 times and count the results? If it is the same model parameters and the same test set, then the results should be the same for each of the 30 tests.

17. Lines 490-493: triclassification and multiclassification issues are elucidated by confusion matrices that illustrate classification mistakes, whereas biclassification is adequately represented by Precision and Recall metrics.

18. Section 4.4 is incomplete, and Fig 16 and Table 6 are seriously missing.

Comments on the Quality of English Language

Moderate editing of English language required.

Author Response

Original Manuscript ID: Sensors-3227472

Original Article Title: An Adaptive Parameter Optimization Deep Learning Model for Energetic Liquid Vision Recognition Based on Feedback Mechanism

Response to Editors and Reviewers

Dear Editors and Reviewers,

We want to show our appreciation to the Editor-in-Chief for giving us this opportunity to revise and resubmit our paper. We also want to thank editors for your time-consuming works and significant contributions. Furthermore, we would take this opportunity to thank the reviewers for their timely and thoughtful comments. Those comments are all valuable and very helpful for revising and improving our paper, as well as the important guiding significance to our researches.

Based on your suggestion and request we have made careful corrected modifications on our manuscript,including revisions to the English language and each comment. Meanwhile, we have uploaded (a) one updated manuscript with red highlighting indicating changes, (b) our point-by-point response to the comments (as below), and (c) a clean updated manuscript without highlights.

Best regards,

Chen Lu, Yang Yuhao, Wu Tianci, Liu Chiang, Li Yang, Tan Jie, Qian Weizhong, Yang Liang, Yue Xiu and Li Gun

1. Reply to Reviewer: 1#

  • Article construction needs to be tweaked. For example, related work section is suggested to be merged into the introduction section. To facilitate comprehension, it is essential to offer materials and methods section, results section, and discussion section. Given the inadequacy of your discussion material, I suggest combining the results section with the discussion section into the results and discussion section.

Author Response:

We would like to thank the reviewer for the valuable and guiding comments. In response to the feedback, we have made significant adjustments to the structure of the article to improve its clarity and coherence. Specifically, we have integrated the "Related Work" section into the "Introduction" and ensure that the background and context are presented in a unified manner.

Additionally, we have significantly expanded the discussion section of the paper and designated it as Chapter 4. The main points of discussion include the following three aspects:

  1. 1. Model Generalization Testing and Discussion
  2. 2. Discussion on the Model Inference Speed in Large-Scale Complex Scenarios
  3. 3 Discussion on the Limitations with the Model and Future Works

We also summarized the current model and system’s shortcomings,and pointed out that further exploration of new techniques, such as transfer learning, is necessary to enhance the model’s learning capacity.The main structure of the paper includes the following four sections:

  1. Introduction
  2. Materials and Proposed Method
  3. Experiments results
  4. Discussion
  5. Conclusion
  • The table's format does not conform to standard criteria; please modify it.

Author Response:

We would like to express our sincere thanks for your constructive and valuable comments. We checked the format of each form and have modified our tables according to the style standard of the official website.Including the newly added tables based on the reviewers' suggestions, the manuscript now contains Tables 1 to 7.

Additionally, we have revised the format of the figures containing subplots according to the journal's template, including Figures 3, 7, 12, 16 and 17.

  • Lines 33-36, Line40, Lines 43-45: references must be included to substantiate the sentences.

Author Response:

We would like to thank the reviewer for the valuable and guiding comments. We have cited relevant references in the corresponding section to substantiate the sentences.Below are the relevant cited references:

  1. Lan Y, Han D, Bai F, et al. Review of research and application of fluid flow detection based on computer vision[C]//Proceedings of the 4th International Conference on Computer Science and Application Engineering. 2020: 1-8.
  2. Ren Z, Fang F, Yan N, et al. State of the art in defect detection based on machine vision[J]. International Journal of Precision Engineering and Manufacturing-Green Technology, 2022, 9(2): 661-691.

  • Line 36 ought to point out the limitations of conventional visual inspection while emphasizing the benefits of visual inspection technologies.

Author Response:

Thank you for your valuable suggestions. Based on your feedback, We point out the limitations of traditional visual inspection techniques and emphasize the advantages of computer vision techniques.It is mentioned at the beginning of the second paragraph in the introduction section,line 37.

While the diversity of test liquids adds complexity to detection, traditional visual inspection often struggles with varying viscosities and densities, leading to inconsistencies and errors. In contrast, computer vision technology can adapt flexibly to the characteristics of energetic liquids, ensuring more accurate measurements.

  • Line 83: please confirm if it is appropriate to cite the 2001 publication that used neural networks for the study here.

Author Response:

Thank you for your valuable feedback and constructive comments. It is indeed inappropriate to refer to the early literature, so we re-consulted the relatively new papers in recent years as a reference.Below are the relevant cited references:

  1. Hosseini S, Taylan O, Abusurrah M, et al. Application of wavelet feature extraction and artificial neural networks for improving the performance of gas–liquid two-phase flow meters used in oil and petrochemical industries[J]. Polymers, 2021, 13(21): 3647.
  • Lines 81-94: recent research progress succinctly enumerate the used methodologies, omitting the outcomes as well as the advantages and disadvantages of the methods.

Author Response:

We would like to thank the reviewer for the valuable and guiding comments. We describe not only the results of each approach, but also their advantages and disadvantages. Wang et al. explored a Coriolis flowmeter model for gas-liquid two-phase flow, utilizing Support Vector Machine (SVM) and Artificial Neural Network (ANN) for flow prediction. By integrating ANN, SVM, and Genetic Programming algorithms, the model's generalization capability and prediction accuracy are enhanced, but it may lead to higher computational costs and take more training Time [15]. Researchers have used wavelet signature extraction and artificial neural network methods to improve the performance of gas-liquid two-phase flowmeters in the oil and gas and petrochemical industries [16]. Drikakis reviewed machine learning applications in fluid dynamics, discussing challenges in liquid viscosity detection and potential future directions [17]. Zepel introduced a general computer vision system for monitoring liquid levels in various chemical experiments [18], while Tim et al. discussed a machine vision system using a single camera for automatic liquid level detection. This method utilizes only a single camera for liquid level detection to reduce hardware costs. It also features a high degree of automation and strong adaptability. However, the algorithm's complexity is high, and the method is affected by the transparency and reflectivity of the liquid [19]. Dejband's team proposed a Deep Neural Networks (DNN) model for accurate water level identification, which offers high accuracy but may require a large amount of training data [20]. Lee et al. applied a Recurrent Neural Network - Long Short-Term Memory (RNN-LSTM) model to real leakage data for feature extraction and classification, providing excellent performance in time-series data but being computationally intensive [21]. He et al. used a Genetic Algorithm - Convolutional Neural Network (GA-CNN) model for real-time prediction of liquid level fluctuations in continuous casting molds, which allows for effective optimization but may be complex to implement [22]. Lin et al. present an enhanced YOLO algorithm for detecting floating debris in waterways. The improvements focus on optimizing detection speed and accuracy. The method excels in real-time performance and debris classification but faces challenges in dealing with overlapping objects and complex backgrounds [23].

  • Section 3.1: covers the Bilateral Filtering Algorithm and the Adaptive Contrast Enhancement (ACE) Algorithm; Section 3.2 covers the DBN Algorithm, LSSVM Algorithm, and GA Optimization Algorithm. Please cite the reference if it is not the original algorithm.

Author Response:

We would like to express our sincere thanks for your constructive and valuable comments. The DBN algorithm, LSSVM algorithm and GA optimization algorithm are really not our original algorithms, It is the foundational module of our algorithm. We have included references to illustrate them.Below are the relevant cited references:

  1. Hua Y, Guo J, Zhao H. Deep belief networks and deep learning[C]//Proceedings of 2015 international conference on intelligent computing and internet of things. IEEE, 2015: 1-4.
  2. Youssef Ali Amer A. Global-local least-squares support vector machine (GLocal-LS-SVM)[J]. Plos one, 2023, 18(4): e0285131.
  3. Katoch S, Chauhan S S, Kumar V. A review on genetic algorithm: past, present, and future[J]. Multimedia tools and applications, 2021, 80: 8091-8126.
  • Lines 210-216: this part is the basics of network training and should state which training set was used for pre-training and how many iterations were used for fine-tuning the parameters.

Author Response:

Thank you for your valuable feedback. This part is explained in detail in Chapter 3 Experiment Results and Discussion chapter.The initial values for the parameters in the loss function are set to , while in the fitness function, parameters are . The tolerance parameters for the error between the detected liquid surface pointer results and actual physical measurements are . The dataset for each exposure consists of 640 samples.In the figure below, we present a portion of the dataset and the detection results. The dataset is continuously being collected and refined during the actual system usage.

During the model training process, 70% of the samples are selected for training, and the remaining 30% are used for testing. Considering the reliability and consistency of the results, each dataset was tested 30 times separately.As shown in the figure below, after 30 iterations, the detection error has become very small, which meets the system's accuracy requirements.In actual system production, it meets the zero-sample testing conditions. After extensive testing, we can ensure that the system meets the high standards of accuracy and responsiveness in real-world scenarios.

  • Lines 346-351: Statements defining properties should include references.

Author Response:

Thank you very much for your constructive feedback. In response to your comments, we have added the corresponding reference to the statement defining the attribute as support.Below are the relevant cited references:

  1. Erkaymaz O, Yapici I S, Arslan R U. Effects of obesity on time-frequency components of electroretinogram signal using continuous wavelet transform[J]. Biomedical signal processing and control, 2021, 66: 102398.
  • Lines 368-371: polynomial interpolation subpixel subdivision algorithm, should be added to the cited literature.

Author Response:

Thank you for your valuable feedback and constructive comments. We add relevant literature to the section describing subpixel subdivision algorithms for polynomial interpolation.Below are the relevant cited references:

  1. Wu S, Zeng W, Chen H. A sub-pixel image registration algorithm based on SURF and M-estimator sample consensus[J]. Pattern Recognition Letters, 2020, 140: 261-266.
  2. Xiao Y, Luo Y, Xin Y, et al. Part coaxiality detection based on polynomial interpolation subpixel edge detection algorithm[C]//2020 3rd World Conference on Mechanical Engineering and Intelligent Manufacturing (WCMEIM). IEEE, 2020: 377-380.
  • Section 3.3.2:It could be abbreviated into a single-paragraph content and integrated into the other sections.

Author Response:

We would like to thank the reviewer for the valuable and guiding comments. Section 3.3.2 can indeed be reduced to a single paragraph, since it only involves a simple calculation of the liquid surface cross-section, no further explanation is necessary.We condense into a single paragraph and use as the beginning of the next chapter.After adjustment, it has been incorporated into Section 2.3.2: Reverse Judgment for Liquid Surface Pointer Detection.

  • S1 and S2 mentioned in line 389 are not labelled in Fig 9, please label them clearly.

Author Response:

We would like to express our sincere thanks for your constructive and valuable comments. We have now marked S1 and S2 in Figure 9. Since the actual image is a conical funnel, the diagram is represented from the side of the container, and the circular area is shown as a straight line on the diagram.The circular area represents the cross-sectional area of the liquid at different heights within the conical measurement container. We have also replaced the relevant elements in Figure 1.

  • Lines 418-423, Fig 10: It is recommended that the figure includes images of real equipment combinations, and that camera models are specified in the text.

Author Response:

We would like to thank the reviewer for the valuable and guiding comments.In response to your comments, we have supplemented the images of the real equipment portfolio and explained the specific camera model in the article in Fig 10 as show below.We used the MER2-U3 series digital camera from DaHeng Imaging's Mercury II generation, along with an HN series 20MP 1" fixed-focus lens.

Due to the sensitivity of the project, we are unable to provide photos of the on-site testing equipment. We hope for your understanding. Instead, we have used images of the actual imaging devices and listed the key equipment specifications in the table below.

Table. The main parameters of the camera for the experiment.

No.

Device

Items

Parameter value

1

Digital Camera

(MER2-2000-19U3M/C)

Port

USB3.0

Resolution rate

5496(H) × 3672(V)

Frame rate

10Hz19.6 fps @5496 × 3672

Sensor

1", Sony IMX183

Rolling shutter CMOS

Pixel size

2.4 μm × 2.4 μm

Spectrum

Black and white/Color

Exposure time

12μs~1s

2

Len

(HN-3522-20M-C1/1X)

Focal length

35mm

Maximum aperture ratio

1:2.2

Maximum imaging size

12.8 x 9.6(Φ16)

Aperture range

F2.2 ‐F16.0

Working distance

0.2m ‐Inf.

Field of view Angle(D*H*V°)

1”

25.8 x 20.8 x 15.7

2/3”

17.9 x 14.4 x 10.8

1/1.8”

14.7 x 11.8 x 8.9

Rear focal length

9.7 mm

  • Doesn't Deep Learning Model Detection connect to the image input in Fig 11, and why doesn't Get Camera Video Images connect to other modules?

Author Response:

Thank you for your valuable suggestions. The question you raised is indeed critical, and we regretfully acknowledge that we made an oversight in this aspect. We inadvertently committed this error due to a lack of attention to detail in the initial submission. In response, we have taken careful steps to correct this by connecting the deep learning model inspection directly to the image input module. This ensures that the entire flowchart is coherent and the procedural steps are fully integrated. We have now revised the framework to ensure that all components of the system function seamlessly together, providing a clear and complete representation of the workflow.

  • Lines 464-469: this should be described here together with the significant data shown in Table 2.

Author Response:

Thank you for your valuable feedback. After Table 2, we have summarized the above content and the important data shown in the table. Our results indicate that the 4-layer RBM model achieves an optimal balance between maintaining high detection accuracy and minimizing inference time and resource consumption. The inference time of the four-layer RBM model is 1.1 milliseconds, the accuracy rate is 99.2%, and the memory usage is 150MB. This makes it the optimal choice for liquid surface pointer detection, ensuring both high detection precision and enhanced efficiency in resource utilization.

  • Lines 476-481: 1. Does the total of 640 sample images include all three exposures, or does it represent 640 images for each exposure? Isn't this data quantity insufficient for a deep learning study such that the outcomes of several model performance evaluations mostly focus on 90% or higher?
  1. Is the allocation of the training and test sets consistently 30% for the test set and 70% randomly designated for the training set? Is there a problem if the training set including the test set?
  2. When testing different models, should the model be tested 30 times with the same model parameters but different test sets, or should the training and test sets be split each time to train 30 times and count the results? If it is the same model parameters and the same test set, then the results should be the same for each of the 30 tests.

Author Response:

Thank you very much for your constructive feedback. In response to your comments, We go into more detail in this section. 1. Each exposure has 640 images, for a total of about 2,000 images. Due to project sensitivity and practical application, this number of data sets is sufficient to meet the needs of this particular scenario. Because the scene of this case is relatively simple, we added a ring light source to keep the detection environment constant. Due to the particularity of the detection target, the accuracy and robustness are highly required, so we use the neural network based on the feedback mechanism to achieve the above requirements. In the follow-up study, we added the laser real-time ranging of liquid level and combined with visual detection to improve the accuracy of the whole system.

Additionally, in different industrial applications, we found that during the initial stage, collecting effective samples is relatively challenging, as there are many redundant or invalid samples, with the number of useful samples typically limited to the scale of several thousand. However, as the system continues to be used, we are consistently collecting and training new data.

With a small dataset, our detection accuracy has indeed exceeded 90%, which is a technical requirement for the system. Under zero-sample production conditions, our detection accuracy needs to reach over 99%.

  1. 70% of each dataset is the training set and 30% is the test set, and the training set will not include the test set data. Therefore, our testing conclusion was derived under zero-sample conditions.

3.When different models are tested, only the model proposed in this paper adopts the parameters described in this paper, and the other models are not modified. The training set is separate from the test set, and each training is performed 30 times.

It is important to note that all tests were conducted in a consistent environment. Due to the small sample size, we performed multiple tests to avoid any accidental results. The variation between each test result was minimal, and we calculated the average to derive our conclusions.

Finally, to validate the system’s performance on larger datasets, we conducted additional experiments during the revision process. Specifically, we tested on the CPPD and EP datasets, which primarily consist of densely packed objects with varying lighting conditions (in contrast to the constant lighting condition in this paper). The entire datasets were used for training, while the system was directly tested under zero-sample production conditions to observe its accuracy. The dataset information is provided below.

Datasets(ourselves)

Images

Objects

Describe

CPPD

16,848

151,632

Various types of medicine boxes made of various materials, covering all mainstream types of pharmacies, including challenges such as reflection caused by waterproof plastic film.

EP

25,127

60,393

A comprehensive sample covering all types of packages in the logistics and express delivery industry, with sizes ranging from 5 centimeters to 3 meters and heights ranging from 0.5 millimeters to 1.2 meters in various shapes.

CPPD

EP

The test results are shown in the figure below. Although the overall performance is not as strong as models specifically designed for certain solutions, the detection accuracy under zero-sample production conditions reached over 95%. This indirectly demonstrates that the model possesses a certain degree of generalization capability.

  • Lines 490-493: triclassification and multiclassification issues are elucidated by confusion matrices that illustrate classification mistakes, whereas biclassification is adequately represented by Precision and Recall metrics.

Author Response:

Thank you for your valuable feedback and constructive comments. While Precision and Recall can provide information about the accuracy and coverage of positive class predictions, they do not directly represent the classification performance of negative classes. The confusion matrix can provide a more comprehensive view of the performance of the Negative class classification by showing the number of True Negative (TN) and False Positive (FP), thereby helping to gain a deeper understanding of the global performance of the model.

  • Section 4.4 is incomplete, and Fig 16 and Table 6 are seriously missing.

Author Response:

We would like to thank the reviewer for the valuable and guiding comments. In the first uploaded file, Figure 16 and 17 are seriously missing due to version conversion issues. We reconfirmed that they were present properly.

Additionally, we have refined Figure 16 by adding a schematic of the external hardware structure to help readers better understand the system architecture, as shown in the figure below.

(a) The Actual Detection Images of the Liquid Surface Pointer.

(b) The Hardware System Architecture.

Figure 16. The Actual Detection Images of the Liquid Surface Pointer and Hardware System Architecture.

Figure 17. Actual Detection Results (Detection Samples)

Response to Comments on the Quality of English Language

Comments on the Quality of English Language:Moderate editing of English language required.

Author Response:

Thank you for your suggestion. We have improved the English expression and grammar typos errors and tone of the whole paper.

Reviewer 2 Report

Comments and Suggestions for Authors

I have encountered some points in the paper that seem unclear or potentially misunderstood. I would like to clarify these aspects by commenting as follows

1.    It appears that the abstract and conclusion sections are quite similar. I would recommend revising these sections to differentiate their purposes more clearly.

2.    The model achieves an inference speed of 1.5 ms/frame. How does this speed hold up when scaled to larger datasets or more complex liquid environments?

3.   Could you please clarify how the inference speed was calculated? A detailed explanation of the methodology used to measure this would be helpful.

4.   Can the authors explain the mechanics behind the reverse judgment algorithm in more detail? How does the feedback mechanism impact the optimization of model parameters, and how does it differ from traditional feedback models?

5.   Bilateral filtering preserves edges while reducing noise. How does this specifically help in detecting small objects such as the pointer tip on the liquid surface? Are there any limitations or edge cases where this approach might fail?

6.   How are the camera's imaging settings (e.g., resolution, frame rate) optimized for detecting the small pointer tip under low-light conditions? Would different camera configurations impact the detection performance of the model?

7.   The paper suggests that varying lighting conditions affect contrast. Has the model been tested under different lighting scenarios (e.g., extreme low-light or bright conditions)? What is the model’s performance under challenging environments?

8.   How scalable is the DBN-AGS-FLSS model to different liquid types or varying industrial applications? Can the model adapt to other types of fluid dynamics beyond what was tested in the paper?

9.   What are the limitations of the current model, and how could it be improved in future work? Are there any plans to address potential weaknesses like robustness under extreme conditions or integration with real-time monitoring systems?

10.         Given the increase in computational complexity with the addition of RBM layers, how do the authors manage resource utilization and training time in the real-time detection of liquid surface pointers? Is there a specific threshold for the number of layers beyond which the model’s efficiency is compromised?

Please provide further explanation and incorporate additional details into the paper where necessary to improve clarity, based on the comments I have provided. 

Author Response

Original Manuscript ID: Sensors-3227472

Original Article Title: An Adaptive Parameter Optimization Deep Learning Model for Energetic Liquid Vision Recognition Based on Feedback Mechanism

Response to Editors and Reviewers

Dear Editors and Reviewers,

We want to show our appreciation to the Editor-in-Chief for giving us this opportunity to revise and resubmit our paper. We also want to thank editors for your time-consuming works and significant contributions. Furthermore, we would take this opportunity to thank the reviewers for their timely and thoughtful comments. Those comments are all valuable and very helpful for revising and improving our paper, as well as the important guiding significance to our researches.

Based on your suggestion and request we have made careful corrected modifications on our manuscript,including revisions to the English language and each comment. Meanwhile, we have uploaded (a) one updated manuscript with red highlighting indicating changes, (b) our point-by-point response to the comments (as below), and (c) a clean updated manuscript without highlights.

Best regards,

Chen Lu, Yang Yuhao, Wu Tianci, Liu Chiang, Li Yang, Tan Jie, Qian Weizhong, Yang Liang, Yue Xiu and Li Gun

1. Reply to Reviewer: 2#

  • It appears that the abstract and conclusion sections are quite similar. I would recommend revising these sections to differentiate their purposes more clearly.

Author Response:

We would like to express our sincere thanks for your constructive and valuable comments. We have revised the summary and the conclusion to make a clearer distinction between their respective content and purpose. The revised content is as follows:

Abstract: The precise detection of liquid flow and viscosity is a crucial challenge in industrial processes and environmental monitoring due to the variety of liquid samples and the complex reflective properties of energetic liquids. Traditional methods often struggle to maintain accuracy under such conditions. This study addresses the complexity arising from sample diversity and the reflective properties of energetic liquids by introducing a novel model based on computer vision and deep learning. We propose the DBN-AGS-FLSS, an integrated deep learning model for high-precision, real-time liquid surface pointer detection. The model combines Deep Belief Networks (DBN), Feedback Least Squares SVM classifiers (FLSS), and Adaptive Genetic Selectors (AGS). Enhanced by bilateral filtering and adaptive contrast enhancement algorithms, the model significantly improves image clarity and detection accuracy. The use of a feedback mechanism for reverse judgment dynamically optimizes model parameters, enhancing system accuracy and robustness. The model achieved an accuracy, precision, F1-score, and recall of 99.37%, 99.36%, 99.16%, and 99.36%, respectively, with an inference speed of only 1.5 ms/frame. Experimental results demonstrate the model's superior performance across various complex detection scenarios, validating its practicality and reliability. This study opens new avenues for industrial applications, especially in real-time monitoring and automated systems, and provides valuable reference for future advancements in computer vision-based detection technologies.

Conclusion: This research successfully presents an innovative approach for detecting liquid flow and viscosity, essential for various industrial and environmental applications. By leveraging advanced computer vision and deep learning techniques, the proposed DBN-AGS-FLSS model demonstrates high precision in identifying liquid surface pointers in real time. Additionally, the implemented feedback mechanism for reverse judgment effectively fine-tunes the model parameters, resulting in enhanced accuracy and reliability. The model was tested on three datasets and compared with DNN, GA-CNN, RNN-LSTM, and YOLOv10. Through ablation experiments, we identified the optimal number of layers for the model, balancing system performance and efficiency. Iterative testing verified the effectiveness of the system's adaptive parameter tuning mechanism, ultimately keeping the error within 0.1%. Based on the above research, we confirming its effectiveness across diverse detection scenarios. Finally, to validate the generalization capability of the model, we conducted zero-sample experiments in other industrial scenarios and datasets. We also discussed potential future research directions and optimizations in this field for further exploration.This work provides a reference for future advancements in liquid detection technologies, emphasizing the importance of continuous improvement and interdisciplinary collaboration in this field.

  • The model achieves an inference speed of 1.5 ms/frame. How does this speed hold up when scaled to larger datasets or more complex liquid environments?

Author Response:

Thank you for raising this important question. Our model currently achieves an inference speed of 1.5 ms per frame due to the optimized algorithmic efficiency and the use of appropriate hardware in our experimental setup(GPU-accelerated performance). Our calculation of inference speed is based on the input of a single image into the model, and the average is taken after multiple tests. We did not perform batch input of large-scale data to calculate the real-time frame rate.

When considering scalability to larger datasets or more complex liquid environments, we have taken the following factors into account to ensure the model can maintain its speed:

1.Hardware Optimization: The inference speed is also highly dependent on the hardware used. In industrial applications, our model can be deployed on optimized hardware platforms such as GPUs or TPUs. This ensures that even when handling more complex data, the inference time remains within an acceptable range.

2.Parallelization and Inference Optimization: To further enhance the inference speed, we plan to implement parallel computing techniques in future work. Methods such as distributed inference or pipeline inference can boost efficiency, especially in environments with multiple cameras or large-scale monitoring systems. By optimizing data flow processing, we can maintain real-time performance even with increased complexity.

3.Feature Simplification and Adaptive Adjustment: In more complex liquid environments, where differences in liquid reflectivity and viscosity may present additional challenges, we can employ adaptive feature extraction algorithms. This allows us to simplify unimportant features and ensure that the feature extraction process in complex environments does not significantly slow down the inference speed.

  • Could you please clarify how the inference speed was calculated? A detailed explanation of the methodology used to measure this would be helpful.

Author Response:

We would like to thank the reviewer for your affirmation and recognition. We are processing individual images, not video streams or batches, and using GPU acceleration. The inference time was tested under consistent environment and hardware conditions. By inputting a single frame of image into the model and performing multiple tests, we calculated the average inference time. Specifically, the process involves recording the time when the single image frame is fed into the network, and then recording the time again when the inference result is generated. The difference between these two times represents the inference time. It is usually done after the model has been trained to evaluate the performance of the model in a test or application scenario.

In a deep learning model, the calculation of inference speed can be achieved through the timing module in Python code. The timer records the time when the inference starts and ends, and the inference time is the difference between the two. Time is usually measured in seconds, and to convert it to milliseconds, you need to multiply it by 1000. We also calculate the inference speed for other models with the same hardware and the same Settings. The following is the output of the program:

  • Can the authors explain the mechanics behind the reverse judgment algorithm in more detail? How does the feedback mechanism impact the optimization of model parameters, and how does it differ from traditional feedback models?

Author Response:

Thank you for your reminder and suggestions. This mechanism involves three key steps:

1.Geometric Parameter Measurement: The liquid forms a circular region as it descends in the container. The radius and area of this circle are detected using computer vision techniques like edge detection and sub-pixel refinement. Using the known relationship between the geometry of the container and the liquid level, the actual area and height of the liquid can be computed.

Reverse Judgment: Once the geometric parameters are calculated, the detected liquid surface area is compared with the expected area based on the physical model. If the detected value closely matches the computed value, the detected pointer is considered valid and accurate. This ensures the reliability of the detection process.

2.Feedback Mechanism: The reverse judgment result is then fed back into the model's loss function, particularly affecting the weight adjustments of the DBN-AGS-FLSS model. By incorporating this feedback, the model dynamically adjusts the SVM classification boundaries, ensuring more accurate classification in the future.The loss function of FLSS and the hyperparameter matrix in AGS were adjusted based on the error, as shown in the formula below. For more details, please refer to Sections 2.2.2 and 2.2.3 of the revised manuscript.

3.Traditional feedback models typically adjust based on the classification error alone, without integrating domain-specific geometric verification. The reverse judgment mechanism used in this study stands out because it includes a domain-specific correction process based on the specificity of the detection structure, which ensures that the feedback is not just based on classification performance but also on physical accuracy. This additional layer of verification makes the model more reliable, especially in scenarios with noisy or incomplete data, such as reflections on liquid surfaces or varying densities.

  • Bilateral filtering preserves edges while reducing noise. How does this specifically help in detecting small objects such as the pointer tip on the liquid surface? Are there any limitations or edge cases where this approach might fail?

Author Response:

Thank you for your thorough review and feedback. Bilateral filtering adjusts pixel weights by considering both geometric proximity and brightness similarity. The pointer typically has a high contrast, especially relative to the liquid background, allowing bilateral filtering to maintain the sharp edges of the pointer while removing background noise, ensuring the accurate detection of the pointer tip.

However, if the contrast between the pointer and the liquid surface is low, particularly under complex lighting conditions or reflective liquid surfaces, bilateral filtering may exhibit limitations. In such cases, the brightness similarity factor may not effectively distinguish the pointer from the background, leading to edge blurring. To overcome these limitations, we subsequently combined an image enhancement technique: Adaptive Contrast Enhancement (ACE), which further enhances the detection of small objects. ACE improves the contrast between the pointer and the background, compensating for the potential shortcomings of bilateral filtering under complex lighting environments.

  • How are the camera's imaging settings (e.g., resolution, frame rate) optimized for detecting the small pointer tip under low-light conditions? Would different camera configurations impact the detection performance of the model?

Author Response:

We would like to thank the reviewer for the valuable and guiding comments. Different camera configurations will have a certain impact, we have compared the same price of flat cameras, their performance is not very different, more important is to determine the camera's imaging Settings. To ensure the consistency of the dataset, it is essential to maintain uniform imaging quality, especially when working with small samples, as varying image quality can impact the model's performance. To avoid this issue, besides ensuring high imaging quality, it is necessary to expand the dataset when switching to different cameras. This is an ongoing effort for us. Currently, with the system in production, we are continuously collecting and expanding the training set by increasing both the number and diversity of samples.

After repeated testing and deployment, we determined the optimal imaging Settings of the camera. Below is our debugging page:

we used the MER2-U3 series digital camera from DaHeng Imaging's Mercury II generation, along with an HN series 20MP 1’’ fixed-focus lens. The key equipment specifications in the table below.

Table. The main parameters of the camera for the experiment.

No.

Device

Items

Parameter value

1

Digital Camera

(MER2-2000-19U3M/C)

Port

USB3.0

Resolution rate

5496(H) × 3672(V)

Frame rate

10Hz19.6 fps @5496 × 3672

Sensor

1", Sony IMX183

Rolling shutter CMOS

Pixel size

2.4 μm × 2.4 μm

Spectrum

Black and white/Color

Exposure time

12μs~1s

2

Len

(HN-3522-20M-C1/1X)

Focal length

35mm

Maximum aperture ratio

1:2.2

Maximum imaging size

12.8 x 9.6(Φ16)

Aperture range

F2.2 ‐F16.0

Working distance

0.2m ‐Inf.

Field of view Angle(D*H*V°)

1”

25.8 x 20.8 x 15.7

2/3”

17.9 x 14.4 x 10.8

1/1.8”

14.7 x 11.8 x 8.9

Rear focal length

9.7 mm

  • The paper suggests that varying lighting conditions affect contrast. Has the model been tested under different lighting scenarios (e.g., extreme low-light or bright conditions)? What is the model’s performance under challenging environments?

Author Response:

We would like to express our sincere thanks for your constructive and valuable comments. Due to the sensitivity and practical application of the case, the accuracy and robustness are very high, we added a ring light source to keep the detection environment constant. In the follow-up study, we added real-time laser liquid level ranging, combined with visual detection, to improve the accuracy of the entire system. However, in order to verify the generalization of our model and its performance in different environments, we have carried out experimental tests in low-illumination environments. The renderings are as follows, which can prove that the performance of our model is excellent.

  • How scalable is the DBN-AGS-FLSS model to different liquid types or varying industrial applications? Can the model adapt to other types of fluid dynamics beyond what was tested in the paper?

Author Response:

Thank you for your valuable feedback. The DBN-AGS-FLSS model proposed in this paper is suitable for the measurement of different kinds of energetic liquids. In the actual system usage, we are detecting various formulations and types of energetic liquids. From the perspective of visual detection, our primary focus is on detecting specific probes in the context of different fluid backgrounds. The energetic liquids we mainly deal with include nitrate ester liquids, nitric acid-based liquids, liquid rocket propellants, and water gas, among others.

To validate the system’s performance on larger datasets and varying industrial applications, we conducted additional experiments during the revision process. Specifically, we tested on the CPPD and EP datasets, which primarily consist of densely packed objects with varying lighting conditions (in contrast to the constant lighting condition in this paper). The entire datasets were used for training, while the system was directly tested under zero-sample production conditions to observe its accuracy. The dataset information is provided below.

Datasets

(By ourselves)

Images

Objects

Describe

CPPD

16,848

151,632

Various types of medicine boxes made of various materials, covering all mainstream types of pharmacies, including challenges such as reflection caused by waterproof plastic film.

EP

25,127

60,393

A comprehensive sample covering all types of packages in the logistics and express delivery industry, with sizes ranging from 5 centimeters to 3 meters and heights ranging from 0.5 millimeters to 1.2 meters in various shapes.

CPPD

EP

The test results are shown in the figure below. Although the overall performance is not as strong as models specifically designed for certain solutions, the detection accuracy under zero-sample production conditions reached over 95%. This indirectly demonstrates that the model possesses a certain degree of generalization capability.

  • What are the limitationsof the current model, and how could it be improved in future work? Are there any plans to address potential weaknesses like robustness under extreme conditions or integration with real-time monitoring systems?

Author Response:

Thank you very much for your constructive feedback. We have added this section to Chapter 4, the discussion chapter, The specific content is as follows:

Current model limitations:

1.Dependence on Preprocessing: The effectiveness of the model relies heavily on the accuracy of the initial image preprocessing (bilateral filtering and ACE). If the preprocessing steps fail to adequately enhance the image or remove noise, it could affect the overall detection performance. 

2.Hardware and Depth Perception Issues: While the system compensates for depth perception issues, variations in hardware setup (such as camera positioning) can introduce additional errors, which the model might struggle to fully account for.

3.Currently, the detection and correction feedback rely solely on visual input. To enhance the reliability of the system, it would be advisable to incorporate more robust mechanisms such as laser ranging or 3D cameras, which could serve as additional safety measures.

Potential future improvements: 

1.Advanced Deep Learning Models: Future research could involve the incorporation of more advanced deep learning algorithms and architectures, such as transformer-based models, to improve the robustness of the model under challenging conditions like extreme lighting or reflection 

2.Integration with Real-Time Monitoring Systems: The current model shows promise in real-time liquid monitoring, but further optimizations could ensure even faster inference times, making it more compatible with industrial applications where real-time decision-making is critical.

3.Improving Robustness in Extreme Conditions: There is potential for enhancing the model's robustness by introducing additional feedback loops or hybrid algorithms that can dynamically adjust the model parameters during extreme conditions, such as variable lighting or reflective surfaces. This could help maintain high accuracy regardless of environmental changes.

4.Incorporating laser ranging for real-time detection of liquid surface height, combined with multi-sensor fusion, can significantly improve the model's performance by providing more accurate corrections and enhancing overall system reliability.

  • Given the increase in computational complexity with the addition of RBM layers, how do the authors manage resource utilization and training time in the real-time detection of liquid surface pointers? Is there a specific threshold for the number of layers beyond which the model’s efficiency is compromised?

Author Response:

Thank you for your valuable feedback and constructive comments. The increase in the number of RBM layers leads to a significant increase in computational complexity, training time, and resource consumption. With the increase of the number of layers, although the feature extraction ability of the model is enhanced, the accuracy is improved, but the inference speed and memory consumption will increase simultaneously. To further reduce the training time, we use an unsupervised pre-training phase, which allows the model to converge quickly with less labeled data. This also effectively reduces the overall training time and improves resource efficiency. We believe that there is a threshold on the number of RBM layers, that is, when the number of layers exceeds a certain value, the inference time and resource occupancy increase disproportionately, despite the improved accuracy of the model, and may lead to overfitting. In the experiments in this paper, we observed that 4-layer RBM is the optimal choice because it strikes an optimal balance between performance and resource utilization. Further increases in the number of layers (such as 5 or more) provide some performance gains, but the inference time is significantly increased, as is the resource consumption. Therefore, we decided to stop increasing the model complexity at 4 layers of RBM. The related ablation experiments are shown in the table below:

No.

RBM Layers

Inference Time (ms)

Accuracy (%)

Memory Usage (MB)

A

2

0.7

98.7

90

B

3

0.9

98.9

120

C

4

1.1

99.2

150

D

5

1.5

99.3

190

E

6

2.1

99.5

240

Response to provide further explanation and incorporate additional details

Please provide further explanation and incorporate additional details into the paper where necessary to improve clarity, based on the comments I have provided.

Author Response:

Thank you for your suggestion. Based on the comments you provided, we have explained parts of the article in more detail and mentioned more information where necessary to improve clarity.

We have addressed the issues you raised in the manuscript, and the content of the paper has been expanded from 23 pages to 28 pages.

Reviewer 3 Report

Comments and Suggestions for Authors

An adaptive parameter optimization deep learning model for energetic liquid vision recognition based on a feedback mechanism is presented in this research. The authors have chosen a good topic; however, the following observations may help improve the quality of the publication. 

1. In the abstract, including a few lines about the background research problem may enhance the reader's interest. It is suggested to use the full forms followed by their abbreviations. Additionally, highlighting the significance of the research for the scientific community at the end of the abstract would add value. 

2. The introduction section can be expanded for better clarity and context.

 3. The workflow diagram of the entire study (i.e., Figure 1) needs to be explained in a more comprehensive manner.

 4. The organization of the article should be presented at the end of the introduction section.

 5. The related work section is too brief. Each related work should be explained separately, and a tabulated presentation would be helpful for comparing the proposed work with existing studies.

 6. The proposed model appears to be well-structured. However, some figures are enclosed within boxes, while others are not. Ensure consistency in formatting.

 7. For Figure 4, it would be more logical if the input is placed at the top and the output at the bottom of the figure.

 8. In Figure 5, it is suggested to place the start point at the top of the figure, and the decision box should not include the word "Stop." Please use a more appropriate term.

 9. The vertical scale is missing in Figure 15. Please include it.

 10. In Figure 16, there appears to be some missing information. Please review and correct it.

 11. The discussion section should be moved to a separate section before the conclusion. Additionally, the conclusion needs to be revised as it currently resembles the abstract too closely.

Comments on the Quality of English Language

Minor English corrections is needed. 

Author Response

1. Reply to Reviewer: 3#

  • In the abstract, including a few lines about the background research problem may enhance the reader's interest. It is suggested to use the full forms followed by their abbreviations. Additionally, highlighting the significance of the research for the scientific community at the end of the abstract would add value.

Author Response:

Thank you for your suggestion and recognition. We have included some background research questions in the abstract. In addition, the significance of this study for the scientific community is highlighted at the end of the abstract: The precise detection of liquid flow and viscosity is a crucial challenge in industrial processes and environmental monitoring due to the variety of liquid samples and the complex reflective properties of energetic liquids. Traditional methods often struggle to maintain accuracy under such conditions. Besides, this study opens new avenues for industrial applications, especially in real-time monitoring and automated systems, and provides valuable reference for future advancements in computer vision-based detection technologies.

We provided a separate explanation of the full name for DBN-AGS-FLSS in the abstract later on. Deep Belief Networks (DBN), Feedback Least Squares SVM classifiers (FLSS), and Adaptive Genetic Selectors (AGS).

  • The introduction section can be expanded for better clarity and context.

Author Response:

We would like to express our sincere thanks for your constructive and valuable comments. According to your comments, we expanded the introduction to make the content of the article clearer and more relevant to the context.The main revisions we made to the Introduction section include the following:

  1. We integrated the content of the "Related Work" section into the Introduction and expanded the discussion of related research, highlighting their key features.
  2. We added explanations to Figure 1 to clarify the scope of the study.
  3. We elaborated on the limitations of traditional vision-based detection algorithms.
  4. We summarized the algorithms from related studies, along with their characteristics and application scenarios, in a table.
  5. The organization of the article presented at the end of the introduction section.
  • The workflow diagram of the entire study (i.e., Figure 1) needs to be explained in a more comprehensive manner.

Author Response:

Thank you for your detailed and constructive comments. In response to your comments, we have made a comprehensive summary of the workflow flow chart in Figure 1, which is as follows: The detection model proposed in this paper firstly obtains real-time images of the liquid surface through a network camera, and then performs image filtering and enhancement. Then the liquid surface pointer is effectively detected by DBN-AGS-FLSS deep learning model. And a reverse judgment algorithm based on computer vision geometric parameter measurement is used to verify the detection results and feed the reverse judgment results into the deep learning model for parameter optimization. Finally, the flow rate and viscosity of the liquid are calculated by recording the time of the upper and lower Pointers on the liquid surface and calculating the time difference.

  • The organization of the article should be presented at the end of the introduction section.

Author Response:

We would like to express our sincere thanks for your constructive and valuable comments. We have corrected the problem you mentioned,We placed the organization and contributions of this paper at the end of the Introduction. First, we introduce the significance of the research and the existing problems, followed by an analysis of related studies. Finally, we present the proposed solution, the structure, and the contributions of this paper. This logical structure provides a coherent and complete flow.

  • The related work section is too brief. Each related work should be explained separately, and a tabulated presentation would be helpful for comparing the proposed work with existing studies.

Author Response:

We would like to express our sincere thanks for your constructive and valuable comments. We have supplemented the related work section, each related work has added their advantages and disadvantages, and presented them in table form. Wang et al. explored a Coriolis flowmeter model for gas-liquid two-phase flow, utilizing Support Vector Machine (SVM) and Artificial Neural Network (ANN) for flow prediction. By integrating ANN, SVM, and Genetic Programming algorithms, the model's generalization capability and prediction accuracy are enhanced, but it may lead to higher computational costs and take more training Time [15]. Researchers have used wavelet signature extraction and artificial neural network methods to improve the performance of gas-liquid two-phase flowmeters in the oil and gas and petrochemical industries [16]. Drikakis reviewed machine learning applications in fluid dynamics, discussing challenges in liquid viscosity detection and potential future directions [17]. Zepel introduced a general computer vision system for monitoring liquid levels in various chemical experiments [18], while Tim et al. discussed a machine vision system using a single camera for automatic liquid level detection. This method utilizes only a single camera for liquid level detection to reduce hardware costs. It also features a high degree of automation and strong adaptability. However, the algorithm's complexity is high, and the method is affected by the transparency and reflectivity of the liquid [19]. Dejband's team proposed a Deep Neural Networks (DNN) model for accurate water level identification, which offers high accuracy but may require a large amount of training data [20]. Lee et al. applied a Recurrent Neural Network - Long Short-Term Memory (RNN-LSTM) model to real leakage data for feature extraction and classification, providing excellent performance in time-series data but being computationally intensive [21]. He et al. used a Genetic Algorithm - Convolutional Neural Network (GA-CNN) model for real-time prediction of liquid level fluctuations in continuous casting molds, which allows for effective optimization but may be complex to implement [22]. Lin et al. present an enhanced YOLO algorithm for detecting floating debris in waterways. The improvements focus on optimizing detection speed and accuracy. The method excels in real-time performance and debris classification but faces challenges in dealing with overlapping objects and complex backgrounds [23].

Model

Summarize

Pros and Cons

Application

Artificial Neural Networks (ANN)

Mimicking the way neurons in the brain work

Non-linear mapping, generalization ability.

Requires large datasets, Lack of interpretability.

Image recognition, speech recognition, medical diagnosis

Deep Neural Networks (DNN)

Deep neural networks composed of multiple hidden layers.

High accuracy, scalability.

Requires large amounts of data, overfitting.

Image recognition, speech recognition, natural language processing.

Recurrent Neural Network - Long Short-Term Memory (RNN-LSTM)

A recurrent neural network that processes time-series data.

Strong in handling time-series data, ability to retain information.

Computationally expensive, slow training time.

Speech recognition, financial time series forecasting, and leak detection in time-series data.

Genetic Algorithm - Convolutional Neural Network (GA-CNN)

Using GA to optimize CNN parameters, leading to improved performance on specific tasks.

Strong optimization capability, automated parameter selection.

Complex implementation, Slow convergence.

Real-time monitoring and optimization scenarios.

YOLO (You Only Look Once)

A real-time object detection algorithm that processes the entire image at once.

Real-time detection, high speed and accuracy.

Struggles with small object detection, difficulty with overlapping objects.

Autonomous driving, video surveillance, traffic monitoring, and real-time object detection in various fields.

  • The proposed model appears to be well-structured. However, some figures are enclosed within boxes, while others are not. Ensure consistency in formatting.

Author Response:

We would like to thank the reviewer for the valuable and guiding comments.  Based on your comments, we made sure that all the models and pictures were consistent in format.We also have revised the format of the figures containing subplots according to the journal's template, including Figures 3, 7, 12, 16 and 17.

  • For Figure 4, it would be more logical if the input is placed at the top and the output at the bottom of the figure.

Author Response:

We would like to express our sincere thanks for your constructive and valuable comments. We put the input at the top of the graph and the output at the bottom, which is really more logical. Below is the modified image:

  • In Figure 5, it is suggested to place the start point at the top of the figure, and the decision box should not include the word "Stop." Please use a more appropriate term.

Author Response:

Thank you for your valuable feedback. We put the start point at the top of the diagram, removed the "Stop" word, and normalized the decision box terminology. Below is the revised image:

  • The vertical scale is missing in Figure 15. Please include it.

Author Response:

Thank you for your reminder, and we apologize for the issue caused by our oversight. We have redrawn the figure and added the missing scales, as show below.

  • In Figure 16, there appears to be some missing information. Please review and correct it.

Author Response:

We would like to thank the reviewer for the valuable and guiding comments. In the first uploaded file, Figure 16 and 17 are seriously missing due to version conversion issues. We reconfirmed that they were present properly.

Additionally, we have refined Figure 16 by adding a schematic of the external hardware structure to help readers better understand the system architecture, as shown in the figure below.

(a) The Actual Detection Images of the Liquid Surface Pointer.

(b) The Hardware System Architecture.

Figure 16. The Actual Detection Images of the Liquid Surface Pointer and Hardware System Architecture.

Figure 17. Actual Detection Results (Detection Samples)

  • The discussion section should be moved to a separate section before the conclusion. Additionally, the conclusion needs to be revised as it currently resembles the abstract too closely.

Author Response:

We would like to thank the reviewer for the valuable and guiding comments. We have separated the discussion section from the results section as a separate paragraph. Both the summary and the conclusion sections have been revised to prevent them from being too similar. The following is the revised conclusion:

Conclusion: This research successfully presents an innovative approach for detecting liquid flow and viscosity, essential for various industrial and environmental applications. By leveraging advanced computer vision and deep learning techniques, the proposed DBN-AGS-FLSS model demonstrates high precision in identifying liquid surface pointers in real time. Additionally, the implemented feedback mechanism for reverse judgment effectively fine-tunes the model parameters, resulting in enhanced accuracy and reliability. The model was tested on three datasets and compared with DNN, GA-CNN, RNN-LSTM, and YOLOv10. Through ablation experiments, we identified the optimal number of layers for the model, balancing system performance and efficiency. Iterative testing verified the effectiveness of the system's adaptive parameter tuning mechanism, ultimately keeping the error within 0.1%. Based on the above research, we confirming its effectiveness across diverse detection scenarios. Finally, to validate the generalization capability of the model, we conducted zero-sample experiments in other industrial scenarios and datasets. We also discussed potential future research directions and optimizations in this field for further exploration.This work provides a reference for future advancements in liquid detection technologies, emphasizing the importance of continuous improvement and interdisciplinary collaboration in this field.

Additionally, we have significantly expanded the Discussion section of the paper and designated it as Chapter 4. The main points of discussion include the following three aspects:

  1. 1. Model Generalization Testing and Discussion
  2. 2. Discussion on the Model Inference Speed in Large-Scale Complex Scenarios

4.3 Discussion on the Limitations with the Model and Future Works

We also summarized the current model and system’s shortcomings,and pointed out that further exploration of new techniques, such as transfer learning, is necessary to enhance the model’s learning capacity.The main structure of the paper includes the following four sections:

Response to Comments on the Quality of English Language

Minor English corrections is needed.

Author Response:

Thank you for your suggestion. We have improved the English expression and grammar of the whole paper.

Reviewer 4 Report

Comments and Suggestions for Authors

The authors propose the integrated deep-learning model for high-precision, real-time liquid surface pointer detection. The model combines Deep Belief Networks, Feed-back Least Squares SVM classifiers, and Adaptive Genetic Selectors. Enhanced by bilateral filtering and adaptive contrast enhancement algorithms, the model significantly improves image clarity and detection accuracy. Using a feedback mechanism for reverse judgment dynamically optimizes model parameters, enhancing system accuracy and robustness. Experimental results demonstrate the model's superior performance across various complex detection scenarios, validating its practicality and reliability.

The paper presents a good scientific level, and I recommend publication.

Author Response

1. Reply to Reviewer: 4#

Comments to the Author

The authors propose the integrated deep-learning model for high-precision, real-time liquid surface pointer detection. The model combines Deep Belief Networks, Feed-back Least Squares SVM classifiers, and Adaptive Genetic Selectors. Enhanced by bilateral filtering and adaptive contrast enhancement algorithms, the model significantly improves image clarity and detection accuracy. Using a feedback mechanism for reverse judgment dynamically optimizes model parameters, enhancing system accuracy and robustness. Experimental results demonstrate the model's superior performance across various complex detection scenarios, validating its practicality and reliability.

The paper presents a good scientific level, and I recommend publication.

Author Response:

We would like to thank the reviewer for your affirmation and recognition. We also appreciate your valuable time. Moving forward, we will continue to maintain a diligent work attitude and carefully proofread the manuscript.

Round 2

Reviewer 2 Report

Comments and Suggestions for Authors

Thank you for your response. After receiving it, the details of the paper became much clearer.